# Characterizing functional pathways of the human olfactory system

**Guangyu Zhou[1]\*, Gregory Lane[1], Shiloh L Cooper[1], Thorsten Kahnt[1,2], Christina Zelano[1]\***

[1]Department of Neurology, Feinberg School of Medicine, Northwestern University, Chicago, United States; [2]Department of Psychology, Weinberg College of Arts and Sciences, Northwestern University, Evanston, United States

**Abstract** The central processing pathways of the human olfactory system are not fully understood. The olfactory bulb projects directly to a number of cortical brain structures, but the distinct networks formed by projections from each of these structures to the rest of the brain have not been well-defined. Here, we used functional magnetic resonance imaging and k-means clustering to parcellate human primary olfactory cortex into clusters based on whole-brain functional connectivity patterns. Resulting clusters accurately corresponded to anterior olfactory nucleus, olfactory tubercle, and frontal and temporal piriform cortices, suggesting dissociable whole-brain networks formed by the subregions of primary olfactory cortex. This result was replicated in an independent data set. We then characterized the unique functional connectivity profiles of each subregion, producing a map of the large-scale processing pathways of the human olfactory system. These results provide insight into the functional and anatomical organization of the human olfactory system.

DOI: https://doi.org/10.7554/eLife.47177.001

**\*For correspondence:**
guangyu.zhou@northwestern.edu
(GZ);
c-zelano@northwestern.edu (CZ)

**Competing interest:** See
page 18

**Reviewing editor:** Joel
Mainland, Monell Chemical
Senses Center, United States

## Introduction

The human sense of smell serves a variety of important functions in everyday life (*Bushdid et al., 2014*; *Devanand et al., 2015*; *McGann, 2017*). It is used to monitor the safety of inhaled air (*Pence et al., 2014*) and edibility of food (*Yeomans, 2006*). It also strongly impacts our social and emotional lives (*Durand et al., 2013*; *Endevelt-Shapira et al., 2018*; *Frumin et al., 2015*; *Gelstein et al., 2011*; *Krusemark et al., 2013*; *Walla, 2008*; *Walla et al., 2003*). Thus, the brain must extract different types of information from odor stimuli, including information about the identities of objects and foods, environmental hazards, and social and emotional cues. These functions are likely carried out by distinct cortical networks within the olfactory system, yet the organization of these functional networks is not fully understood.

This incomplete understanding is due partly to ambiguity about the anatomical and functional properties of the cortical targets of human olfactory bulb projections. Collectively, these areas are commonly referred to as primary olfactory cortex (*Carmichael et al., 1994*; *Feher and Feher, 2017*; *Gottfried, 2010*; *Mai and Paxinos, 2012*; *Price, 2009*) (although see *Wilson, 2009*; *Haberly, 2001*; *Chapuis and Wilson, 2011* and others for discussions of the accuracy of this definition of the pirmary olfactory cortex). In humans, this includes the anterior olfactory nucleus, the olfactory tubercle, the frontal and temporal piriform cortices, and subregions of both the amygdala and entorhinal cortex (*Allison, 1954*; *Eslinger et al., 1982*; *Gonçalves Pereira et al., 2005*; *Insausti et al., 2002*; *Milardi et al., 2017*). The fact that the olfactory bulb simultaneously projects directly to a number of structures suggests parallel functional pathways in the olfactory system (*Haberly, 2001*; *Kauer, 1991*), but the distinct roles of these primary olfactory areas and their functional pathways are not fully understood (*Bensafi et al., 2007*; *Gottfried et al., 2006*; *Gottfried et al., 2004*;

*Gottfried et al., 2002*; *Howard et al., 2009*; *Li et al., 2008*; *Li et al., 2006*; *Sobel et al., 2000*; *Sobel et al., 1999*; *Zelano et al., 2005*). Additionally, the olfactory system is organized differently than other sensory systems, which contain pre-cortical thalamic relays, further suggesting a deeper understanding of the organization of olfactory networks in the human brain is warranted.

The vast majority of research on primary olfactory cortex has focused on piriform cortex, which is the largest recipient of bulbar projections. Most of this research has been conducted in rodents, where piriform cortex is divided into anatomically and functionally distinct anterior and posterior subdivisions (*Calu et al., 2007*; *Grau-Perales et al., 2019*; *Haberly and Price, 1978*; *Stettler and Axel, 2009*; *Yang et al., 2017*). In humans, the anatomy and functionality of piriform cortex is less understood. Although it can be divided into frontal- and temporal-lobe subregions (*Mai et al., 2015*; *Vaughan and Jackson, 2014*; *Young et al., 2018*; *Allison, 1954*), whether these correspond to rodent anterior and posterior subdivisions is unclear. While neuroimaging studies have pointed to functional heterogeneity within human piriform cortex (*Bensafi, 2012*; *Fournel et al., 2016*; *Gottfried et al., 2002*; *Howard et al., 2009*; *Howard and Gottfried, 2014*; *Li et al., 2008*; *Porter et al., 2005*; *Seubert et al., 2013*; *Zelano et al., 2011*; *Zelano et al., 2005*), its anatomical and functional distinctions are still not clearly defined.

While numerous rodent and human studies have focused on piriform cortex, far fewer have examined other primary olfactory structures, such as the anterior olfactory nucleus and the olfactory tubercle. These structures have been anatomically well-defined in rodents (*Aqrabawi and Kim, 2018a*; *Haberly and Price, 1978*; *Shipley and Adamek, 1984*), primates (*Carmichael et al., 1994*), and humans (*Allison, 1954*; *Eslinger et al., 1982*; *Mai et al., 2015*), but their roles in olfactory processing are not fully understood in any of these species (*Gadziola et al., 2015*; *Wesson and Wilson, 2011*). Recent rodent data suggest that the anterior olfactory nucleus may be involved in odor memory (*Aqrabawi and Kim, 2018b*; *Oettl et al., 2016*) and localization (*Kikuta et al., 2010*), and the olfactory tubercle may play an important role in multisensory integration and attention (*Wesson and Wilson, 2010*; *Zelano et al., 2005*), although a complete understanding of the functions of these areas is lacking.

Previous studies have used task-related and resting functional magnetic resonance imaging (fMRI) to examine olfactory networks, using primary olfactory and orbitofrontal cortices as seed regions (*Banks et al., 2016*; *Cecchetto et al., 2019*; *Fjaeldstad et al., 2017*; *Karunanayaka et al., 2017*; *Karunanayaka et al., 2014*; *Kiparizoska and Ikuta, 2017*; *Kollndorfer et al., 2015*; *Krusemark and Li, 2012*; *Nigri et al., 2013*; *Sreenivasan et al., 2017*; *Sunwoo et al., 2015*). These studies have contributed important broad knowledge of parallel olfactory networks (*Karunanayaka et al., 2014*), how they compare to trigeminal networks (*Karunanayaka et al., 2017*), and how they change with age (*Wang et al., 2005*) and disease (*Caffo et al., 2010*; *Fjaeldstad et al., 2017*; *Killgore et al., 2013*; *Sunwoo et al., 2015*; *Wang et al., 2010*; *Wang et al., 2015*). However, the functional connectivity profiles of the primary olfactory subregions have not been considered separately. This is important because these subregions, which receive direct and parallel input from the bulb, likely form the anatomical substrates of ethological, parallel olfactory networks. Therefore, a quantitative characterization of the distinct functional pathways of human primary olfactory subregions would be an important step toward understanding the large-scale networks that underlie the basic, parallel, purposes of olfactory processing. The discovery of unique whole-brain connectivity profiles for the different primary olfactory subregions could also provide insight into the nature of the distinct functions of these areas in olfactory perception. This information, in turn, could have clinical implications for diseases that impact particular primary olfactory subregions.

Thus, the goals of this study were two-fold: first, to test the hypothesis that primary olfactory subregions form distinct large-scale olfactory processing networks; and second, if so, to characterize these networks across the whole brain. For the first, we used well-established, unsupervised k-means clustering techniques (*Eickhoff et al., 2018*; *Kahnt et al., 2012*; *Kahnt and Tobler, 2017*; *Wang et al., 2017*), to parcellate primary olfactory cortex into distinct clusters based solely on whole-brain connectivity patterns. We reasoned that if whole-brain functional connectivity patterns alone could be used to accurately parcellate primary olfactory cortex into its established, anatomically-defined subregions, this would suggest that these subregions do in fact form distinct functional pathways. We also reasoned that if these parcellation results were robust, the results should replicate in an independent data set. For the second, we characterized the distinct functional connectivity patterns of each primary olfactory subregion in order to produce a whole-brain map of the networks

formed by each area. Our results provide insight into the functional and anatomical organization of the human olfactory system and provide a basis for future investigation into the functions of the distinct cortical targets of the olfactory bulb.

## Results

We used resting-state fMRI connectivity to examine the functional pathways of human primary olfactory cortex in two main steps. First, we used k-means clustering techniques to parcellate primary olfactory cortex into distinct clusters. These clusters were based on the group-level, whole-brain functional connectivity of all voxels within primary olfactory cortex, separately for each hemisphere. Second, using the results of the parcellation analysis, we characterized the distinct, large-scale networks of the human olfactory system. To do this, we first determined that there were no hemispheric differences in the connectivity profiles of primary olfactory subregions, suggesting we should combine corresponding clusters across the left and right hemispheres. We then quantified the whole-brain functional networks that were unique to each subdivision, and those that were common to all subdivisions. Finally, as a discussion point, we considered the functional properties of connected brain areas for each primary olfactory subdivision and attempted to form a speculative hypothetical model of human olfactory functional networks.

### Parcellation of human primary olfactory cortex

To test the hypothesis that primary olfactory subregions form distinct, large-scale olfactory networks, we tested whether their anatomical boundaries could be accurately delineated based on whole-brain functional connectivity maps. To do this, we conducted a functional-connectivity-based parcellation of human primary olfactory cortex. Twenty-five subjects (average ± standard error age: 25.5 ± 1.2 years; 14 female) underwent a 10 min resting-state fMRI scan. We first outlined the entirety of primary olfactory cortex into a combined region-of-interest (ROI) on which to perform the k-means clustering analysis. This ROI was drawn for the left and right hemispheres separately, according to a human brain atlas which contains detailed demarcation of most primary olfactory areas (*Mai et al., 2015*; *Ongür et al., 2003*) (*Figure 1A*). The ROI included only those subregions with detailed boundaries in the atlas, and consisted of a combination of the anterior olfactory nucleus, olfactory tubercle, and frontal and temporal piriform cortices (*Figure 1B*), defined based on *Mai et al. (2015)*. Note that additional primary olfactory areas, including amygdala and entorhinal cortex (*Allison, 1954*; *Carmichael et al., 1994*; *Eslinger et al., 1982*; *Gonçalves Pereira et al., 2005*; *Zatorre et al., 1992*), were not included in our ROI because the exact location of olfactory afferents into these areas is poorly understood (*Gonçalves Pereira et al., 2005*). This is an important topic for future investigation of human olfactory networks.

We then estimated the whole-brain functional connectivity profile of each voxel within the ROI by computing the Pearson correlation coefficient between the resting-state fMRI time-series of a given voxel and that of every other voxel in the rest of the brain. This resulted in subject-wise connectivity matrices. We then performed a leave-one-out analysis (*Kahnt et al., 2012*) to estimate the stability of the connectivity profiles of individual primary olfactory cortex voxels across participants, as a prerequisite for averaging the connectivity matrices across subjects (*Figure 2A*). To examine the similarity between the individual functional connectivity matrices, we computed a histogram of correlation values between individual matrices and the group matrix. The histogram of correlation values showed that the similarity of connectivity patterns was above zero in all voxels in primary olfactory cortex (mean correlation coefficient: 0.19, standard error: 0.0041), justifying averaging across subjects. Note that because they are computed across rest-of-the-brain voxels, $R$ values larger than 0.0088 are statistically significant at $p < 0.05$ (Bonferroni corrected for the number of voxels in primary olfactory cortex). To parcellate within-ROI voxels into subdivisions based on their whole-brain functional connectivity profiles, we applied unsupervised k-means clustering methods to the average connectivity matrix. We used a priori k = 4 based on the fact that our ROI was comprised of four anatomically distinct brain regions. For both the left and right hemispheres, this analysis successfully parcellated the primary olfactory cortex ROI into four distinct brain regions that corresponded anatomically to the anterior olfactory nucleus, olfactory tubercle, and frontal and temporal piriform cortices (*Figure 2B*).

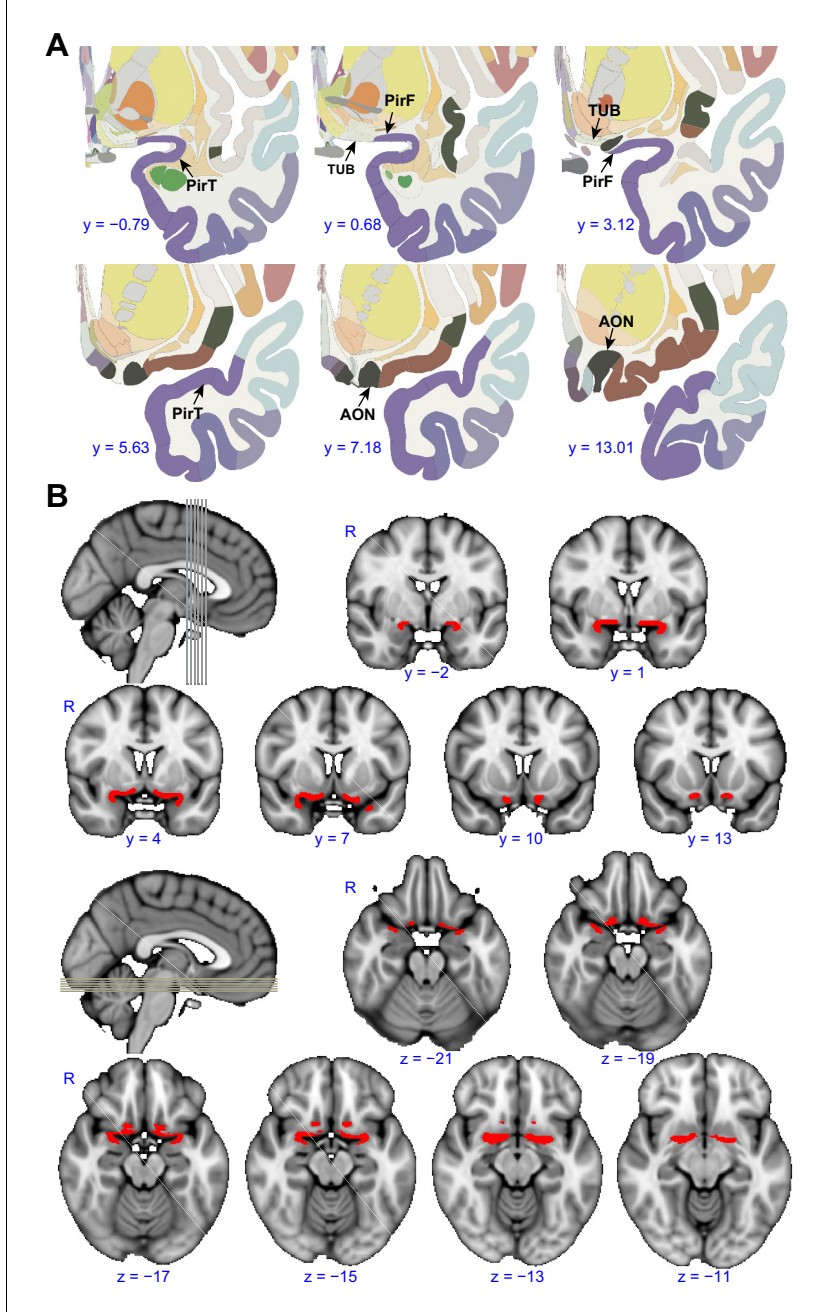

**Figure 1.** Region of interest. (**A**) Panels show examples from the human brain atlas used to define the region of interest used in the parcellation analysis. Relevant areas include the anterior olfactory nucleus (AON), olfactory tubercle (TUB), and frontal (PirF) and temporal (PirT) piriform cortex (**Mai et al., 2015**). (**B**) The region of interest shown overlaid on the FSL's MNI152_T1_1mm_brain. The coronal and axial slices correspond to the vertical and horizontal lines on the sagittal slice respectively. R, right hemisphere.

DOI: https://doi.org/10.7554/eLife.47177.002

The following source data is available for figure 1:

**Source data 1.** Relates to *Figure 1*.

DOI: https://doi.org/10.7554/eLife.47177.003

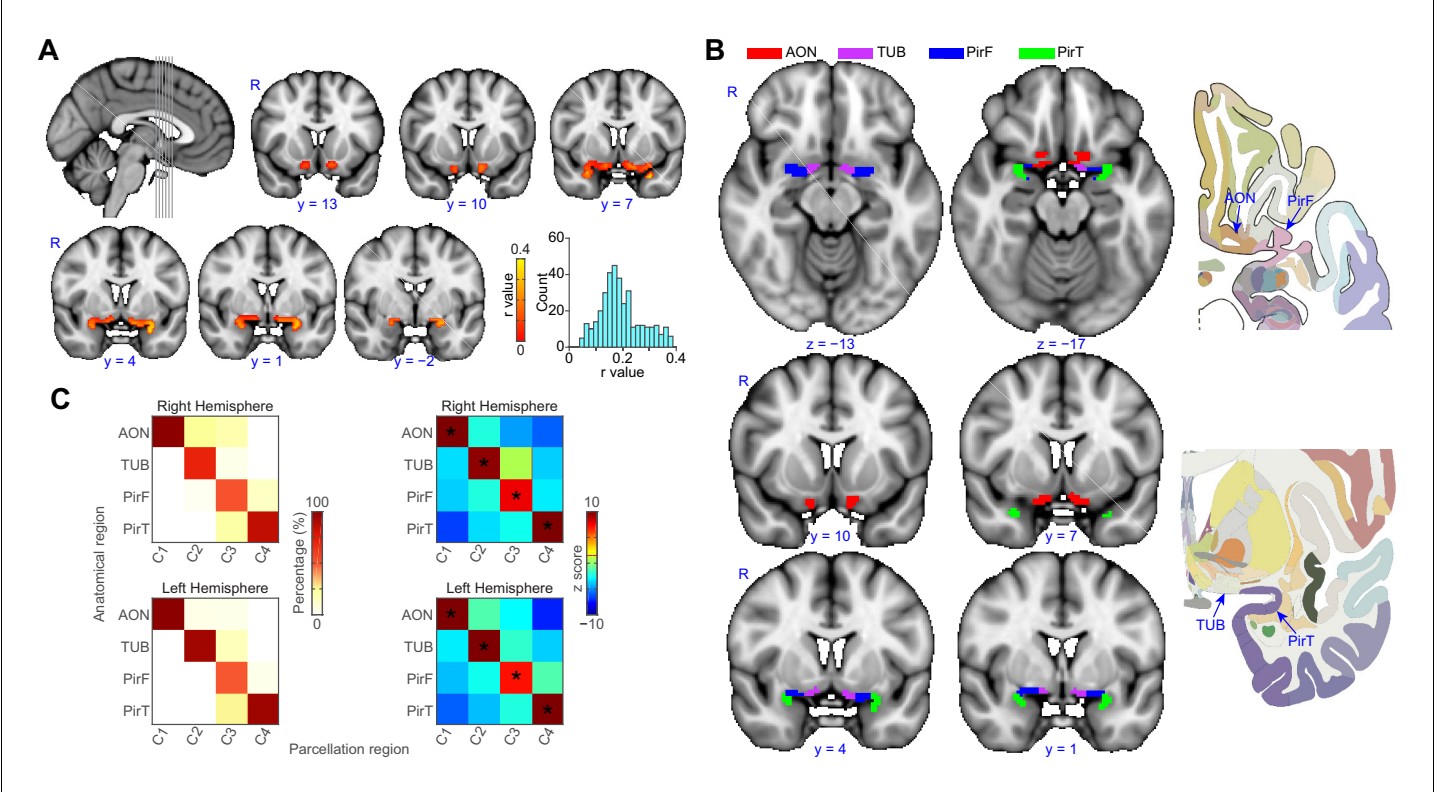

**Figure 2.** Parcellation of human left and right primary olfactory cortex. (**A**) Inter-subject stability of functional connectivity patterns. The correlation of the functional connectivity patterns between each subject and all other subjects was calculated for each voxel using a leave-one-out method. The coronal slices, corresponding to the vertical lines on the sagittal slice, show the average stability map. The bar plot shows the histogram of the correlation values. (**B**) k-means (k = 4) clustering results shown on the FSL's MNI152_T1_1mm_brain. The right column shows one axial and one coronal slice of the Atlas (*Mai et al., 2015*). (**C**) Parcellation accuracy of each subregion. Left column: proportion of voxels from each parcellation subdivision located within each anatomical subregion. Right column: z score of the proportion maps. * indicates p<0.001 (false discovery rate corrected). R, right hemisphere; AON, anterior olfactory tubercle; TUB, olfactory tubercle; PirF, frontal piriform cortex; PirT, temporal piriform cortex.

DOI: https://doi.org/10.7554/eLife.47177.004

The following source data and figure supplements are available for figure 2:

**Source data 1.** Relates to *Figure 2*, panel (**A**).
DOI: https://doi.org/10.7554/eLife.47177.008

**Source data 2.** Relates to *Figure 2*, panel (**B**) and (**C**).
DOI: https://doi.org/10.7554/eLife.47177.009

**Figure supplement 1.** Replication analyses.
DOI: https://doi.org/10.7554/eLife.47177.005

**Figure supplement 1—source data 1.** Relates to *Figure 2—figure supplement 1*.
DOI: https://doi.org/10.7554/eLife.47177.006

**Figure supplement 2.** Example sagittal slice from one fMRI volume for each subject (S1–S25).
DOI: https://doi.org/10.7554/eLife.47177.007

To confirm the correspondence between our parcellation results and the anatomical delineation of primary olfactory subregions in the Atlas of the Human Brain (*Mai et al., 2015*), we computed the proportion of voxels from each parcellation cluster located within each of the atlas-derived subdivisions, drawn prior to performing the parcellation analysis (*Figure 2B*). The statistical significance of this proportion was tested using a permutation test. Specifically, for each permutation, we shuffled the labels of the anatomical subdivision and re-calculated the proportion. This procedure was repeated 10,000 times, resulting in a distribution of permuted proportions for each parcellation cluster. A z score of the actual proportion values was computed by subtracting the average and then dividing by the standard deviation, which was obtained by normal distribution fitting of the permuted data (Matlab's *normfit*). We found that for each parcellated subdivision, there was one

anatomical ROI that contained significantly more voxels than the other anatomical ROIs (*Figure 2C*, minimal z score = 7.18). Thus, we found that the location of voxels within each parcellated subdivision corresponded to a single anatomically-determined ROI. Specifically, the medial-rostral-most parcellated subdivision corresponded to the anterior olfactory nucleus. The adjacent caudal subdivision corresponded to the olfactory tubercle. Within the frontal lobe, the lateral-rostral subdivision corresponded to the frontal piriform cortex, and within the temporal lobe, the caudal-most subdivision corresponded to temporal piriform cortex.

## Replication of parcellation results

To confirm the robustness of our parcellation results, we performed two control analyses aimed at replicating the initial findings. First, we performed the k-means clustering analysis on a different ROI of primary olfactory cortex, drawn independently by one of the co-authors of this paper. Second, we performed the k-means clustering analysis on an independent data set (N = 53), collected for a previous study on a different scanner, with different acquisition parameters and different subjects (*Kahnt and Tobler, 2017*).

In the first control analysis, performed on an independently drawn ROI, we found that k-means clustering still successfully parcellated primary olfactory cortex into four distinct regions, corresponding anatomically to the anterior olfactory nucleus, olfactory tubercle, and frontal and temporal piriform cortices (*Figure 2—figure supplement 1A,B*). In our second control analysis, performed on an independent data set, we found that, again, primary olfactory cortex successfully parcellated into the same four distinct regions (*Figure 2—figure supplement 1C,D*). Importantly, all analysis steps performed on this independent data set were identical to those performed in our initial analysis. These results suggest good reliability of our finding that human olfactory cortex can be accurately parcellated based on whole-brain functional connectivity patterns.

## Parcellation results across hemispheres and k values

Thus far, we demonstrated that both the left and right primary olfactory areas can be accurately subdivided based on their functional connectivity profiles. To further examine differences between the left and right hemispheres and at different k values, we conducted additional parcellation analyses using a single primary olfactory ROI containing all subregions from both hemispheres, at a range of k values. We reasoned that if connectivity patterns were similar across hemispheres for each subregion, then parcellation analysis of this combined ROI should group left and right sides of each primary olfactory subregion, as opposed to grouping, for example, the neighboring subregions on the same hemisphere. We computed this analysis using k values ranging from 3 to 6 (*Figure 3*). We found that for a k value of 3, the parcellation analysis grouped left and right anterior olfactory nucleus and left tubercle as one cluster, left and right frontal piriform cortex and left temporal piriform cortex as a second cluster, and right temporal piriform cortex alone as the third cluster. For a k value of 4, the analysis successfully grouped left and right hemispheres for both piriform subregions, but it grouped left anterior olfactory nucleus with left olfactory tubercle and right anterior olfactory nucleus with right olfactory tubercle, suggesting these two primary olfactory areas have relatively more lateralized connectivity patterns. A k value of 5 grouped the left and right hemispheres of all subregions, with a fifth cluster consisting of only right temporal piriform cortex. Finally, a k value of 6 grouped all subregions across hemispheres except for the anterior olfactory nucleus. These results indicate a clear separation of frontal and temporal piriform cortex for a wide range of cluster solutions and even across hemispheres (*Figure 3A*). All parcellation results showed good agreement with the anatomical subregions (*Figure 3B,C*). Of note, the anterior olfactory nucleus and olfactory tubercle are classified as one subregion for a clustering solution of k = 3. The left and right anterior olfactory nucleus were separated into different subregions for k = 4, 6. These findings suggest stronger lateralization of connectivity patterns for the anterior olfactory nucleus compared to other primary olfactory areas.

## Primary olfactory cortical functional connectivity does not statistically differ between hemispheres

Our next step was to examine the whole-brain functional connectivity profiles of the different primary olfactory subregions. Prior to performing this analysis, we first determined whether to use

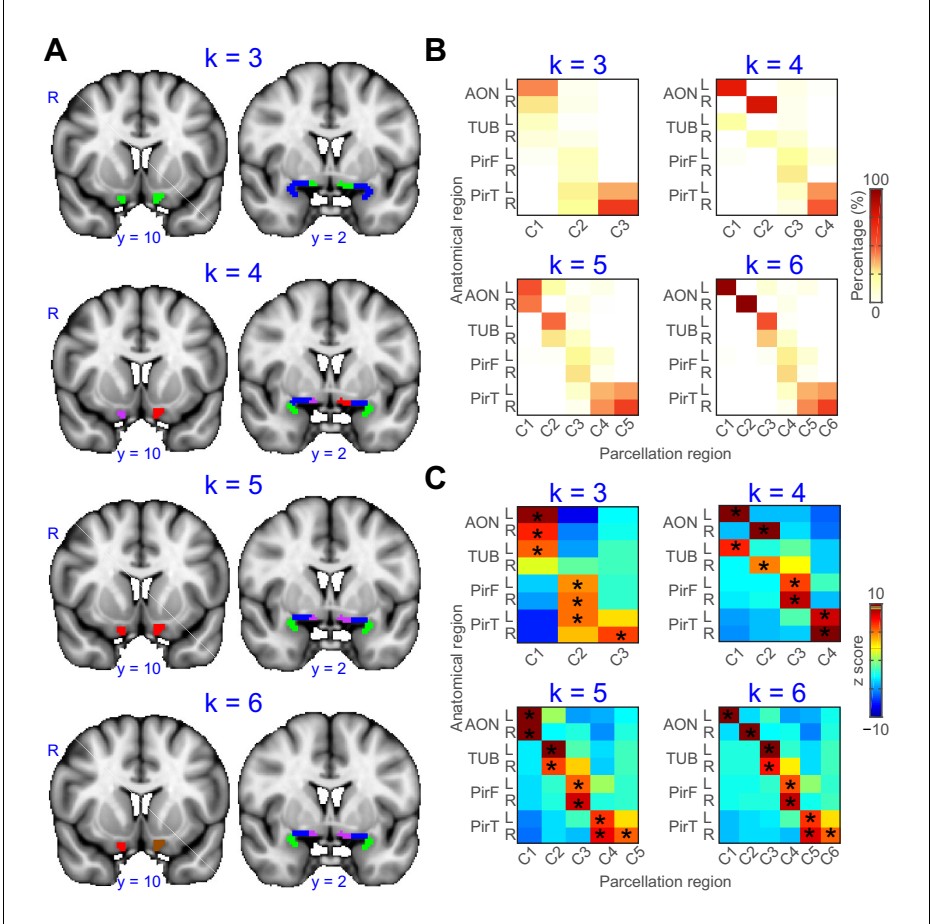

**Figure 3.** Parcellation of primary olfactory cortex combined across left and right hemispheres. (A) k-means clustering results shown on the FSL's MNI152_T1_1mm_brain for k = 3 to 6. Each color represents one cluster. (B) Proportion of voxels of each parcellation subdivision within each anatomical subregion. (C) z score of the proportion maps in panel B. * indicates p<0.001 (false discovery rate corrected). L, left hemisphere; R, right hemisphere; AON, anterior olfactory tubercle; TUB, olfactory tubercle; PirF, frontal piriform cortex; PirT, temporal piriform cortex.

DOI: https://doi.org/10.7554/eLife.47177.010
The following source data is available for figure 3:

**Source data 1.** Relates to *Figure 3*.
DOI: https://doi.org/10.7554/eLife.47177.011

hemispherically combined clusters, or hemispherically distinct clusters. We reasoned that if connectivity profiles of left and right primary olfactory areas did not statistically differ, then they should not be analyzed separately. We therefore conducted a lateralization-index analysis to directly statistically compare connectivity patterns across hemispheres. The lateralization index was defined as ($Z_{\text{left}}$ − $Z_{\text{right}}$)/($Z_{\text{left}}$ + $Z_{\text{right}}$), where $Z_{\text{left}}$ and $Z_{\text{right}}$ were the functional connectivity maps for the left and right seed regions, respectively. We found no statistically significant difference between whole-brain connectivity maps for the corresponding left and right primary olfactory subregions (*Figure 4—figure supplement 1*) (threshold-free cluster enhancement (TFCE) corrected p>0.001). Based on this result, all proceeding analyses were conducted using combined ROIs of corresponding left and right subregions.

## Distinct whole-brain human olfactory networks

The fact that primary olfactory subregions were accurately anatomically parcellated based on their functional connectivity profiles suggests that they form distinct, parallel olfactory networks. To

examine these functional networks, we produced whole-brain maps of the non-overlapping brain areas exhibiting functional connectivity with each subregion. To do this, we first applied a statistical threshold to the whole-brain functional connectivity map for each subregion (TFCE corrected p<0.001), and binarized them (assigned a value of 1 or 0). This resulted in four distinct whole-brain maps. We then further masked them according to whether each voxel exhibited statistically significant functional connectivity with a single subregion or with multiple subregions. This produced two maps of functional connectivity: one with the unique connectivity patterns for each subregion, and the other with connectivity patterns shared by multiple subregions. The complete list of areas showing subregion-specific connectivity is contained in *Table 1*.

Below, we outline the unique connectivity patterns we found for each primary olfactory subregion.

## Functional connectivity profiles of anterior olfactory nucleus and olfactory tubercle

The brain areas that showed connectivity unique to the anterior olfactory nucleus were largest in the orbitofrontal cortex, with smaller clusters in the left inferior temporal gyrus, bilateral anterior temporal gyri, the bilateral anterior insula and the mammillary bodies and retromammillary commissure (*Figure 4A,E*). Areas of connectivity in the orbitofrontal cortex were extensive, including the entire gyrus rectus and encompassing parts of the medial, anterior and lateral orbital gyri. Notably, there was a strong cluster of connectivity corresponding to bilateral areas along the medial orbital sulcus, close to its intersection with the transverse orbital sulcus—this is significant because this part of orbitofrontal cortex is sometimes referred to as human secondary olfactory cortex (*Gottfried and Zald, 2005*). Connectivity with the left inferior temporal gyrus was centered around a posterior region along the established object-recognition pathway. There were also large clusters in the bilateral anterior temporal gyri, and the bilateral anterior insula, in gustatory cortex. Finally, there was a medial cluster centered around the region between the mammillary bodies, the retromammillary commissure and the posterior hypothalamic nucleus.

The brain areas that exhibited connectivity unique to the olfactory tubercle were largest in the medial prefrontal cortex, with smaller clusters in the perisplenial region, the left temporal fusiform cortex, the red nucleus of the brainstem and the accumbens (*Figure 4B,F*). Connectivity between the olfactory tubercle and the medial prefrontal cortex was mainly located in the paracingulate gyrus and frontal pole. In the paracingulate, it centered bilaterally around the anterior-most aspect of Brodman's area 32. Connectivity with the frontal pole was stronger in the left hemisphere, and extended from the dorsal edge of the ventro-medial prefrontal cortex into the medial prefrontal cortex, with a smaller cluster extending even more dorsal, reaching the dorsomedial prefrontal cortex. There was also a cluster of connectivity located bilaterally in the ventral striatum, in the accumbens. There were bilateral clusters of connectivity in both the red nuclei of the brainstem and in the retrosplenial area.

## Functional connectivity profiles of frontal and temporal piriform cortices

Interestingly, the frontal and temporal piriform cortices also showed distinct connectivity patterns, supporting the hypothesis that they are functionally distinct. The frontal piriform cortex showed a large cluster of connectivity with the dorsal striatum (both putamen and caudate), with smaller clusters in the precentral gyrus, the cingulate gyrus, mediodorsal thalamus and left supramarginal gyrus (*Figure 4C,G*). Connectivity with the caudate nucleus and putamen was extensive, covering the bilateral entirety of both regions. In the precentral gyrus, connectivity was evident along the primary motor strip, close to the lateral sulcus, corresponding to the face/lips/tongue/jaw area. There was also a large cluster of connectivity in the caudal anterior (anterior mid-cingulate) cortex, a cluster within the mediodorsal thalamic nucleus extending over the entire anterior-posterior axis, and a cluster unilaterally in the center of the left supramarginal gyrus.

The brain areas that exhibited unique connectivity to the temporal piriform cortex were largest in the bilateral brainstem (centered in the pons) and temporal pole, with smaller clusters in bilateral inferior frontal gyri, bilateral superior temporal gyrus, bilateral hippocampus and bilateral posterior insula (*Figure 4D,H*). In the brainstem, connectivity was maximal in the ventral aspect of the pons

**Table 1.** Summary of functional connectivity results.

| Label | Volume (mm³) | | | | |
|---|---|---|---|---|---|
|  | Overlap | AON | TUB | PirF | PirT |
| Frontal Orbital Cortex | 2592 | 3000 | 144 | - | 1048 |
| Frontal Medial Cortex | 992 | 1120 | - | - | - |
| Cingulate Gyrus | 2760 | 200 | 80 | 2304 | - |
| Insular Cortex | 384 | 496 | - | 224 | 832 |
| Subcallosal Cortex | 3632 | 616 | - | - | - |
| Caudate | 136 | 120 | - | 2024 | - |
| Paracingulate Gyrus | 1336 | - | 2600 | - | - |
| Parahippocampal Gyrus | 296 | - | 464 | - | 2584 |
| Temporal Pole | 328 | - | - | - | 9184 |
| Putamen | 1368 | - | - | 3376 | 96 |
| Hippocampus | 1176 | - | - | 136 | 1448 |
| Amygdala | 2120 | - | - | - | - |
| Accumbens | 336 | - | - | - | - |
| Planum Polare | - | 248 | - | - | 480 |
| Frontal Pole | - | 2792 | 1504 | - | 736 |
| Temporal Fusiform Cortex | - | 688 | 352 | - | 1240 |
| Inferior Frontal Gyrus | - | 208 | - | - | - |
| Inferior Temporal Gyrus | - | 1224 | - | - | 248 |
| Heschl's Gyrus (includes H1 and H2) | - | 80 | - | - | 208 |
| Planum Temporale | - | 104 | - | - | 96 |
| Brainstem | - | - | 592 | - | 6056 |
| Thalamus | - | - | 120 | 1384 | - |
| Pallidum | - | - | - | 504 | - |
| Precentral Gyrus | - | - | - | 1616 | - |
| Postcentral Gyrus | - | - | - | 216 | 296 |
| Frontal Operculum Cortex | - | - | - | 128 | 336 |
| Central Opercular Cortex | - | - | - | 224 | 376 |
| Supramarginal Gyrus | - | - | - | 808 | - |
| Juxtapositional Lobule Cortex | - | - | - | 1104 | - |
| Superior Frontal Gyrus | - | - | - | - | 128 |
| Temporal Occipital Fusiform Cortex | - | - | - | - | 344 |
| Superior Temporal Gyrus | - | - | - | - | 1464 |
| Middle Temporal Gyrus | - | - | - | - | 1368 |
| Angular Gyrus | - | - | - | - | 256 |
| Parietal Operculum Cortex | - | - | - | - | 640 |

The volumes of statistically significant voxels in each brain region are shown for overlapping and subregion-specific clusters. The Overlap column does not include subregion-specific regions. - indicates volume less than 80 mm³ (10 voxels). Atlas query was conducted with FSL's HarvardOxford-cort-maxprob-thr50-2mm and HarvardOxford-sub-max-prob-thr50-2mm atlases. AON, anterior olfactory nucleus; TUB, olfactory tubercle; PirF, frontal piriform cortex; PirT, temporal piriform cortex.

DOI: https://doi.org/10.7554/eLife.47177.012

The following source data is available for Table 1:

**Source data 1.** Functional connectivity profile of each subregion.

DOI: https://doi.org/10.7554/eLife.47177.013

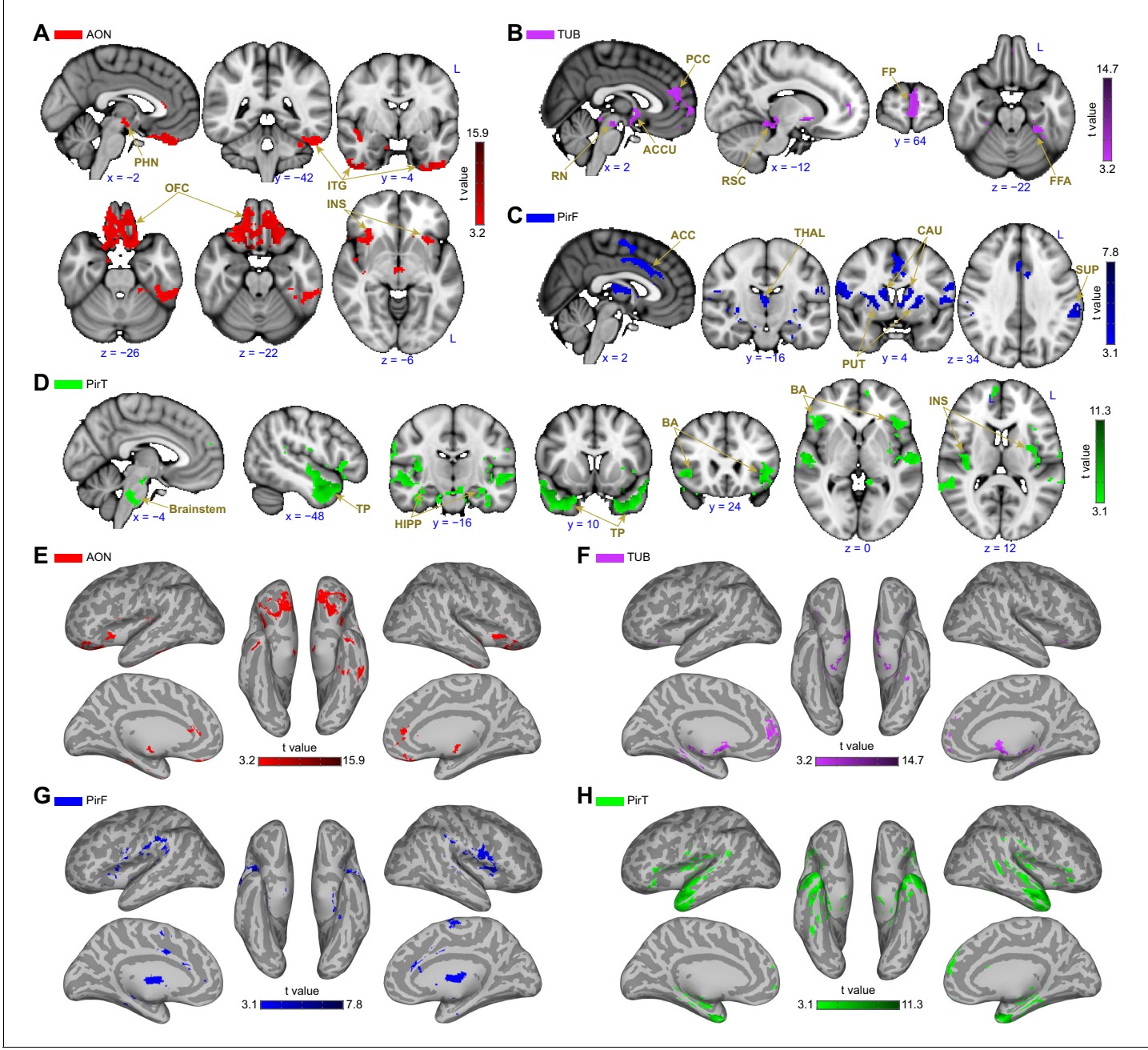

**Figure 4.** Subregion-specific functional connectivity patterns. (A–D) Brain regions that are uniquely positively connected to each of the primary olfactory subregions including the (A) anterior olfactory nucleus (AON, red), (B) olfactory tubercle (TUB, purple), (C) frontal piriform cortex (PirF, blue) and (D) temporal piriform cortex (PirT, green). The results are shown on the FSL's MNI152_T1_1mm_brain. (E–H) Functional connectivity maps shown on Freesurfer's cvs_avg35_inMNI152 brain surface for AON, TUB, PirF and PirT. All functional connectivity maps were thresholded at threshold-free cluster enhancement corrected p<0.001. L, left hemisphere; PHN, posterior hypothalamic nucleus; ITG, inferior temporal gyrus; OFC, orbitofrontal cortex; INS, insula; PCC, paracingulate cortex; RN, retromammillary nucleus; ACCU, accumbens; RSC, retrosplenial area; FP, frontal pole; FFA, fusiform face areas; ACC, anterior cingulate cortex; THAL, thalamus; CAU, caudate; PUT, putamen; SUP, supramarginal gyrus; TP, temporal pole; HIPP, hippocampus; BA, Broca's area.

DOI: https://doi.org/10.7554/eLife.47177.014

The following source data and figure supplements are available for figure 4:

**Source data 1.** Relates to *Figure 4*.
DOI: https://doi.org/10.7554/eLife.47177.017

**Figure supplement 1.** Lateralization of functional connectivity.

*Figure 4 continued on next page*

*Figure 4 continued*

DOI: https://doi.org/10.7554/eLife.47177.015

**Figure supplement 1—source data 1.** Relates to *Figure 4—figure supplement 1*.

DOI: https://doi.org/10.7554/eLife.47177.016

within the nucleus raphe magnus. There were also large clusters of connectivity in the anterior temporal pole. Within the inferior frontal gyrus, connectivity was stronger on the left side within Broca's area, with a smaller cluster in the same area on the right hemisphere. There were clusters in the left superior temporal gyrus, in the language comprehension areas (*Friederici et al., 2003*; *Leff et al., 2009*) and clusters in the right medial temporal gyrus.

## Functional connectivity common to all subregions

Although we have shown that primary olfactory subregions have distinct functional connectivity profiles, they likely also have some functional pathways in common, especially considering the strong reciprocal connectivity between the subregions. To identify the common primary olfactory connectivity network, the connectivity map of each subregion was binarized at a threshold of TFCE corrected p<0.001 to include only those clusters that exhibited connectivity with all subregions. This resulted in a whole-brain connectivity map of areas that are functionally connected to all primary olfactory cortical subregions (*Figure 5*, *Table 1*). We found that these areas included large clusters in the bilateral hippocampus, amygdala and subgenual area, with smaller clusters in the anterior insula and posterior orbitofrontal cortex (*Figure 5A,B*). Connectivity with the hippocampus covered the entire anterior-posterior extent. In the amygdala, connectivity was maximal in the medial subregions and in the central amygdala. Interestingly, for both the right and left primary olfactory cortical subregions, connectivity clusters with the rest of the brain were generally more extensive in the right hemisphere compared to the left hemisphere (*Frasnelli et al., 2010*; *Herz et al., 1999*; *Royet and Plailly, 2004*; *Zatorre et al., 1992*) (*Figure 5A,B*). The common connectivity of all primary olfactory subregions with the orbitofrontal cortex could explain why it is so reliably activated even in basic and passive olfactory tasks.

## Discussion

In the olfactory system, information flows from the bulb to the cortex in a direct and parallel manner. Similarly, other sensory systems have parallel organization of processing pathways, but these typically occur at a later stage of processing. For example, information from other systems, including somatosensation, gustation, vision and audition, is processed in the thalamus prior to primary sensory cortex, and is parallelized downstream from there. In olfaction, thalamic processing occurs *after* direct parallel primary cortical processing, and, based on data from this study, may only occur for a subset of olfactory processing streams. From the olfactory bulb, mitral and tufted cells project to several different brain regions which are thought to play unique roles in olfactory processing (*Ghosh et al., 2011*; *Haberly, 2001*; *Miyamichi et al., 2011*; *Sosulski et al., 2011*). In rodents, the anatomical locations and properties of these regions have been well defined (*Igarashi et al., 2012*; *Nagayama et al., 2010*; *Vassar et al., 1994*). In humans, however, far fewer studies have attempted to identify and define these primary olfactory areas (*Crosby and Humphrey, 1939*; *Eslinger et al., 1982*; *Allison, 1954*; *Shipley and Reyes, 1991*). Similarly, whereas many studies have begun to outline unique functional properties of primary olfactory areas within the rodent olfactory system (*Ikemoto, 2003*; *Illig, 2005*; *Lei et al., 2006*; *Myhrer et al., 2010*; *Wesson and Wilson, 2010*), fewer have done the same in humans. In the current study, we used data-driven connectivity-based parcellation techniques to show that whole-brain functional connectivity patterns alone could be used to parcellate human primary olfactory cortical regions into subregions that anatomically corresponded to the anterior olfactory nucleus, olfactory tubercle and frontal and temporal piriform cortices. Our findings indicate a dissociation in whole-brain functional connectivity patterns across the subregions of human primary olfactory cortex. This suggests that the human olfactory system is comprised of distinct parallel processing pathways (*Cecchetto et al., 2019*; *Savic et al., 2000*) that may be related to the different recipients of projections from the olfactory bulb, in line with previous studies in rodents (*Geramita et al., 2016*; *Haberly, 2001*; *Igarashi et al., 2012*; *Kauer, 1991*;

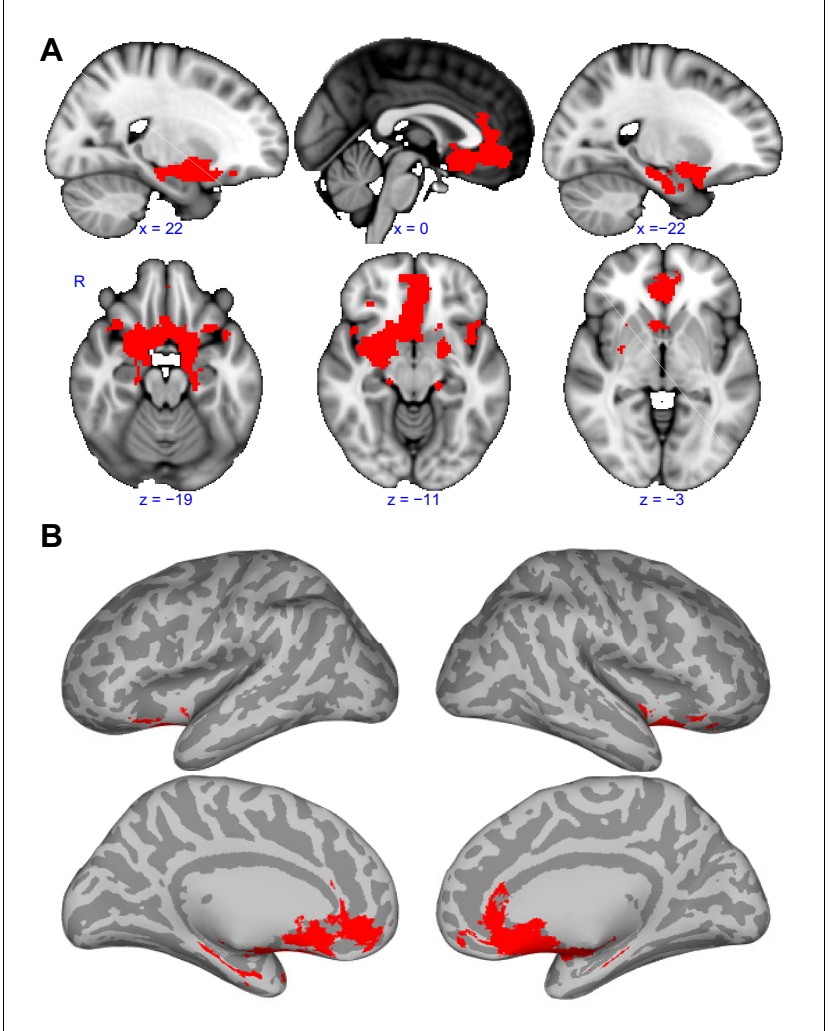

**Figure 5.** Functional connectivity common to all subregions. (**A**) Brain regions that showed statistically significant positive functional connectivity with each of the primary olfactory subregions. Results are overlaid on the FSL's sagittal and axial MNI152_T1_1mm_brain. (**B**) Same brain regions as in (**A**) shown on medial (top row) and lateral (bottom row) Freesurfer's cvs_avg35_inMNI152 brain surfaces. Red indicates the functional connectivity maps that were thresholded at threshold-free cluster enhancement corrected p<0.001. R, right hemisphere.

DOI: https://doi.org/10.7554/eLife.47177.018

The following source data is available for figure 5:

**Source data 1.** Relates to *Figure 5*.

DOI: https://doi.org/10.7554/eLife.47177.019

*Payton et al., 2012*). Our findings provide a detailed description of the particular brain areas that exhibit unique connectivity with each individual primary olfactory subregion. This could be used to gain insight into the specific role that each subregion plays in olfactory perception. We also demonstrate parallel organization of the olfactory system, in which olfactory networks reach a broader set of cortical targets at an earlier stage of processing compared to other sensory systems. This data might have implications for understanding the olfactory decline that appears in early stages of some neurological disease states. For example, olfactory structures that form networks with brain areas implicated in particular pathologies likely perform critical olfactory sensory functions (e.g. the olfactory tubercle is connected to areas involved in depression, see *Croy and Hummel, 2017*, and the temporal piriform cortex is connected to areas involved in primary progressive aphasia, see *Olofsson et al., 2013*).

Across species, the most frequently studied primary olfactory area is piriform cortex. In rodents, it has been consistently shown that projections from the bulb to piriform cortex are spatially distributed (*Ghosh et al., 2011*; *Iurilli and Datta, 2017*; *Miyamichi et al., 2011*) and that information about odor identity can be extracted from the spatiotemporal dynamics of these ensemble patterns (*Haddad et al., 2013*; *Illig and Haberly, 2003*; *Poo and Isaacson, 2009*; *Rennaker et al., 2007*; *Stettler and Axel, 2009*; *Sugai et al., 2005*; *Zhan and Luo, 2010*). At the same time, numerous studies suggest that the function of piriform cortex goes beyond simple odor-identity coding and is strongly impacted by its regional connectivity with other cortical areas (*Cleland and Linster, 2003*; *Sadrian and Wilson, 2015*). Numerous studies have implicated posterior piriform cortex in associative functions such as odor learning and memory (*Calu et al., 2007*; *Chen et al., 2014*; *Choi et al., 2011*; *Gire et al., 2013*; *Gottfried and Dolan, 2003*; *Johnson et al., 2000*; *Karunanayaka et al., 2015*; *Martin et al., 2006*; *Roesch et al., 2007*; *Sacco and Sacchetti, 2010*; *Schoenbaum and Eichenbaum, 1995*), and also suggest that the region may mediate learned olfactory responses and behaviors (*Choi et al., 2011*). Moreover, the strength and composition of piriform networks has also been shown to depend on experience and on the state of the organism (*Chapuis et al., 2013*; *Cohen et al., 2015*; *Cohen et al., 2008*; *Hasselmo and Barkai, 1995*; *Kay and Freeman, 1998*; *Linster and Hasselmo, 2001*; *Wilson and Sullivan, 2011*).

While human studies suggest a role for piriform cortex that goes beyond pure odor-object coding, (*Bensafi, 2012*; *Gottfried, 2010*; *Porter et al., 2007*; *Schulze et al., 2017*; *Zelano et al., 2005*), the anatomical and functional properties of human piriform cortex are not well understood compared to rodents. In fact, in some cases, it is not even clear which human bulbar recipients correspond to the well-defined rodent ones. While it has been suggested that the frontal and temporal subdivisions of human piriform may correspond to the anterior and posterior subdivisions of rodent piriform respectively, there is little anatomical or functional evidence to support this. For example, the rodent anterior piriform receives a greater density of bulb projections than the posterior piriform, while the same is not true for human frontal, relative to temporal, piriform. The lack of clear correspondence between human and rodent piriform subdivisions suggest that functional differences in primary olfactory areas across species could be substantial.

In our study, we found a clear differentiation between the functional connectivity patterns of human frontal and temporal piriform subdivisions, suggesting these two areas play different roles in olfactory processing in the human brain (*Albrecht et al., 2010*; *Bao et al., 2016*; *Bensafi et al., 2007*; *Howard et al., 2009*; *Plailly et al., 2012*; *Porter et al., 2005*; *Zelano et al., 2005*). This distinction between frontal and temporal piriform cortices was robust, surviving across k values and hemispheres. Interestingly, our findings suggest that frontal piriform cortex has strong functional connectivity with motor planning areas, including the caudate/putamen and the primary motor cortex, specifically at the face/nose/jaw section of the motor homunculus (perhaps the facial movement areas include sniffing). Intriguingly, frontal piriform cortex was also connected to the left supramarginal gyrus, an area that has been consistently implicated in tool-grasping in humans (*Glover et al., 2012*; *Johnson-Frey et al., 2005*), leading us to the tempting and highly speculative hypothesis that frontal piriform cortex may play a specific role in combining olfactory information with motor planning in order to guide food with the hand to the mouth. In contrast, we found that the temporal piriform cortex was connected to the brainstem raphe magnus and posterior insula, areas implicated in pain processing (*Segerdahl et al., 2015*; *Woo et al., 2009*) and respiratory modulations (*Ackermann and Riecker, 2010*; *Evans et al., 2009*), as well as the core language network (*Ardila et al., 2014*; *Wible et al., 2005*). Interestingly, in the context of olfaction, respiratory modulation and pain mediation are tightly linked, since many olfactory stimuli also activate trigeminal nerve endings that are located inside the nasal cavities. If a painful stimulus enters the nasal cavities (e.g. ammonia), a protective, fast respiratory reduction occurs, to minimize entry of dangerous chemicals into the body. Intriguingly, verbal communication also requires modulations in breathing, and thus, temporal piriform cortex may mediate both olfactory-related verbal communication and protective changes in respiration. These findings may be applicable to the hypothesized neurocognitive limitations of olfactory language (*Cain, 1979*; *Engen and Ross, 1973*; *Olofsson and Gottfried, 2015*), since our data may suggest that the same olfactory subregion exhibiting connectivity with language networks is involved in other critical olfactory functions.

Beyond piriform cortex, other cortical recipients of olfactory bulb output are less explored, including the anterior olfactory nucleus (*Brunjes et al., 2005*) and olfactory tubercle (*Wesson and*

*Wilson, 2011*). In rodents, the anterior olfactory nucleus is a true cortical structure that can be divided into two main subdivisions, the pars externa and pars principalis (*Pigache, 1970*; *Valverde et al., 1989*). Studies suggest that some of the spatial properties of rodent glomerular activation are preserved in the pars externa, whereas the pars principalis exhibits a spatially distributed activation similar to other primary olfactory areas (*Kay et al., 2011*; *Miyamichi et al., 2011*). This suggests functional heterogeneity across the rodent anterior olfactory nucleus. In humans, the anterior olfactory nucleus is also a true cortical structure, but it does not appear to contain an analog to the pars externa (*Crosby and Humphrey, 1941*). It can be divided anatomically into retrobulbar and cortical anterior and posterior subdivisions (*Mai et al., 2015*; *Ubeda-Bañon et al., 2017*; *Ongür et al., 2003*). That said, only a small number of studies have directly explored the human anterior olfactory nucleus (*Ubeda-Bañon et al., 2017*), so its functional subdivisions remain unclear. Rodent studies suggest the pars externa is involved in odor localization (*Esquivelzeta Rabell et al., 2017*; *Kikuta et al., 2010*) and the pars medialis could be involved in top-down modulation of bulbar responses (*Aqrabawi et al., 2016*). More broadly, rodent studies suggest a role for the anterior olfactory nucleus in the initial formation of representations of odor objects (gestalts) (*Haberly, 2001*; *Lei et al., 2006*), allowing piriform cortex to perform more associative functions, relating information about odor objects to movement and behavior. In our study, we found that whole-brain connectivity of the human anterior olfactory nucleus was maximal in orbitofrontal areas and other regions associated with object recognition. Our findings are in line with rodent studies showing connectivity between anterior olfactory nucleus and orbitofrontal cortex (*Illig, 2005*), and with the hypothesized role for the anterior olfactory nucleus in the formation of odor-object representations (*Haberly, 2001*). We also found that connectivity profiles for the anterior olfactory nucleus were more lateralized than those of other primary olfactory subregions. This finding is in line with rodent studies suggesting that anterior olfactory nucleus neurons can distinguish between signals from the ipsilateral and contralateral nostrils, suggesting representation of lateralized inputs in this region (*Kikuta et al., 2010*).

Similar to the anterior olfactory nucleus, we lack a complete understanding of the function of the olfactory tubercle, especially in humans. The majority of research on the olfactory tubercle has been conducted in rodents and has focused on its relationship with the reward system (*Ikemoto, 2007*), with far fewer studies considering it as a primary olfactory cortical structure (*Wesson and Wilson, 2011*). Interestingly, the olfactory tubercle is the main recipient of tufted (as opposed to mitral) cell projections from the rodent olfactory bulb (*Scott et al., 1980*). Tufted cells show enhanced odor sensitivity, enhanced respiratory entrainment and broader receptive fields compared to mitral cells (*Mori and Shepherd, 1994*; *Shepherd et al., 2004*), suggesting that the olfactory tubercle is important for olfactory tasks requiring high sensitivity. Studies suggest potential roles for the rodent olfactory tubercle in odor discrimination (*Murakami et al., 2005*; *Wesson and Wilson, 2010*), olfactory multisensory integration (*Wang et al., 2010*), state-dependent modulation of olfactory bulb activity (*Gervais, 1979*) and odor reward value (*Howard et al., 2016*). Studies also suggest a role for the rodent olfactory tubercle in sensory hedonics and social behavior (*Gervais, 1979*; *Hagamen et al., 1977*; *Hitt et al., 1973*). Compared to rodents, much less is known about the human olfactory tubercle. In our study, we found connectivity between the olfactory tubercle and brain areas implicated in emotional processing, depression, and social cognition, including anterior paracingulate cortex and left frontal pole (*Bludau et al., 2014*; *Eskenazi et al., 2015*; *Fettes et al., 2018*; *Jackson et al., 2003*; *Koch et al., 2018*; *Koechlin, 2011*; *Papmeyer et al., 2015*; *Veer et al., 2010*). Interestingly, we also found connectivity between the olfactory tubercle and the left fusiform gyrus (fusiform face area), a brain region that is highly responsive to human faces (*Çukur et al., 2013*; *Grill-Spector et al., 2004*; *Kanwisher et al., 1997*; *McCarthy et al., 1997*). Our findings are consistent with rodent data suggesting a role for the tubercle in emotion and social cognition.

In an attempt to summarize and clarify the basic implications of our findings, we created a simplified, speculative illustrative diagram suggesting potential roles for each of the primary olfactory subdivisions that our study parcellated (*Figure 6*). One possible interpretation of our data suggests the following basic functions for human primary olfactory cortical subregions: Anterior olfactory nucleus, a cortical structure, may function as a first step in forming olfactory object representations; The olfactory tubercle may function in extracting social and emotional information from olfactory stimuli, mediating olfactory-related social and emotional responses; Frontal piriform cortex may be involved in planning olfactory-related movements, speculatively those involved in eating; Temporal piriform

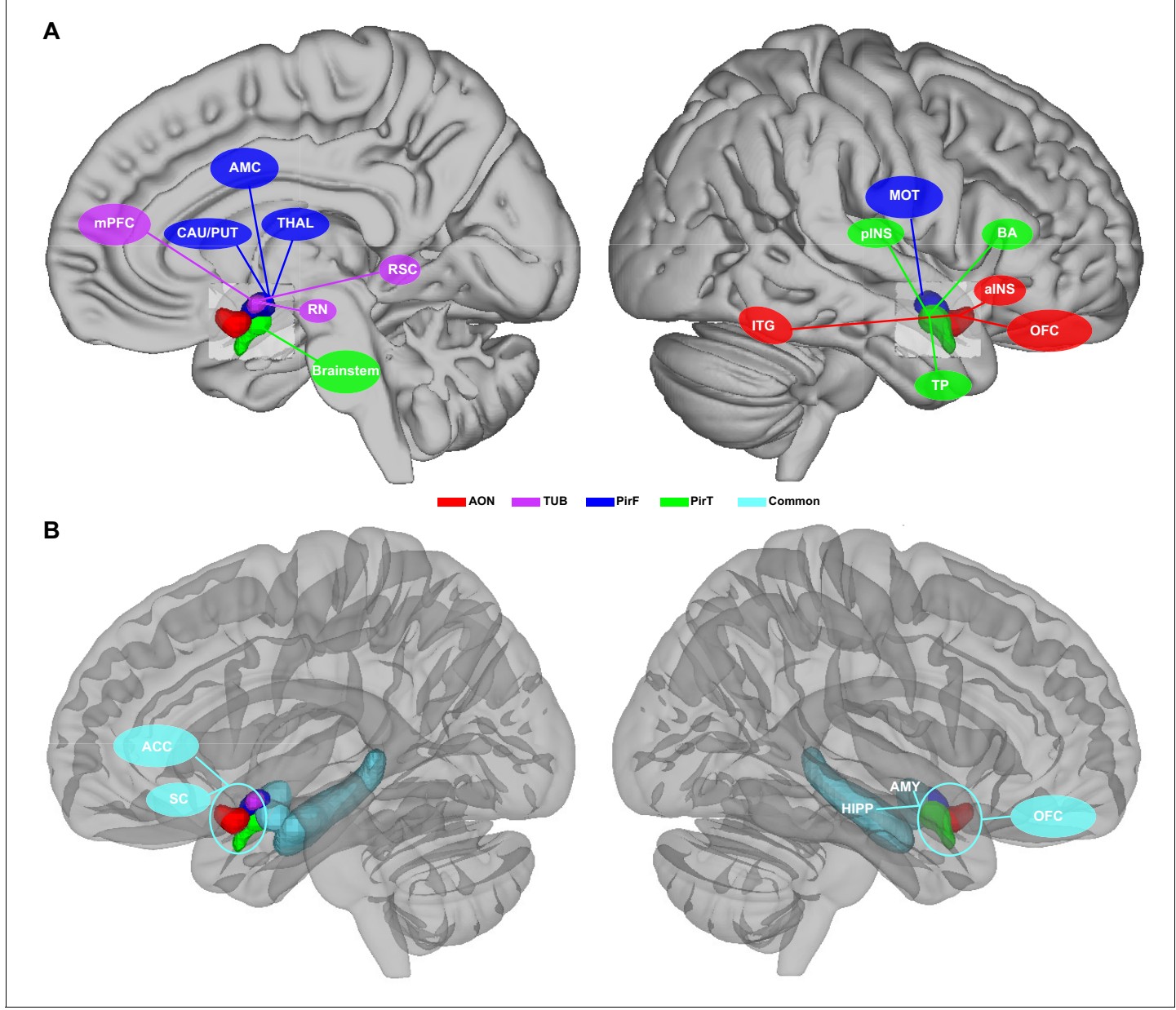

**Figure 6.** Schematic illustrative summary of the functional organization of human primary olfactory cortex. (**A**) Summary of brain regions that are uniquely connected to each subregion, including the anterior olfactory nucleus (AON), olfactory tubercle (TUB), frontal piriform cortex (PirF), and temporal piriform cortex (PirT). (**B**) Summary of brain regions that are commonly connected to all subregions. mPFC, medial prefrontal cortex; CAU, caudate; PUT, putamen; AMC, anterior mid-cingulate cortex; THAL, thalamus; RN, red nucleus; RSC, retrosplenial cortex; ITG, inferior temporal gyrus; pINS, posterior insular cortex; MOT, motor area; TP, temporal pole; BA, Broca's area; aINS, anterior insular cortex; OFC, orbitofrontal cortex; ACC, anterior cingulate cortex; SC, subcallosal cortex; HIPP, hippocampus; AMY, amygdala.

DOI: https://doi.org/10.7554/eLife.47177.020

The following source data is available for figure 6:

**Source data 1.** Relates to *Figure 6*.
DOI: https://doi.org/10.7554/eLife.47177.021

cortex may mediate pain-related (protective) respiratory modulations and verbal communication about olfactory stimuli. Importantly, the functions of primary olfactory areas cannot be determined from resting connectivity networks explored here, and there are numerous alternate interpretations of our data. Future experiments that combine psychophysics with the measurement of neural activity

across primary olfactory areas are needed to gain a full understanding of parallel functional olfactory pathways in the human brain.

## Materials and methods

### Participants

Twenty-five healthy subjects (14 female) with normal or corrected-to-normal vision were recruited for this study. The average (standard error) age was 25.5 (1.2) years. All subjects were right-handed and had normal olfactory function by self-report. No participant reported a history of smell, taste or ear-nose-throat, psychiatry, or neurological disorder. This study was approved by Northwestern University's Institutional Review Board under Protocol #STU00201746. All participants gave their voluntary written consent before the experiment. All experiments were conducted according to the principles of the Declaration of Helsinki.

### MRI data acquisition

Magnetic resonance imaging (MRI) data were acquired on a 3T Siemens TIM Trio scanner equipped with a 64-channel head coil (Siemens Healthcare, Erlangen, Germany), at Northwestern University's Center for Translational Imaging. A 10 min resting fMRI scan was acquired for each subject using a single-shot gradient-echo planar-imaging sequence with following parameters: repetition time (TR): 555 ms; echo time (TE): 22 ms; flip angle: 47°; MB-8 with Split-slice GRAPPA (*Olman et al., 2009*; *Todd et al., 2016*); field of view (FOV): 208 mm; voxel size: $2.0 \times 2.0 \times 2.0$ mm$^3$; 64 axial slices. To reduce the distortion and improve the signal-to-noise ratio in the primary olfactory and orbitofrontal areas, the slice orientation was set to approximately 30° from the AC-PC line (*Deichmann et al., 2003*). These acquisition parameters resulted in robust signals within orbitofrontal and olfactory areas in each subject (*Figure 2—figure supplement 2*). Further, our finding of strong connectivity between orbitofrontal cortex and all primary olfactory cortical areas suggests good orbitofrontal coverage. Subjects were instructed to look at a white fixation cross on a black background and to breathe in and out through their nose.

A high-resolution anatomical image was acquired for each subject using T1-weighted MPRAGE (TR: 2300 ms; TE: 2.94 ms; flip angle: 9°; FOV: 256 mm; voxel size: $1.0 \times 1.0 \times 1.0$ mm$^3$; 176 sagittal slices).

### MRI data preprocessing

The structural images were skull-stripped and segmented into gray matter, white matter and cerebrospinal fluid using the BET (*Smith, 2002*) and FAST (*Zhang et al., 2001*) tools of FSL (FMRIB Software Library, www.fmrib.ox.ac.uk/fsl; RRID:SCR_002823) (*Jenkinson et al., 2012*; *Smith et al., 2004*; *Woolrich et al., 2009*). The resulting white matter and cerebrospinal fluid images were further eroded by one voxel (FSL's *fslmaths*).

Preprocessing of the resting fMRI data included removal of the first 10 volumes, motion correction and generating spatial registration matrices using FSL's FEAT. Each subject's functional images were normalized to their anatomical image using the brain-boundary registration method, and each individual anatomical image was registered to the Montreal Neurological Institute (MNI) standard brain (MNI152_T1_2mm_brain) using the non-linear registration method (12 degrees of freedom). Linear and quadratic trends were removed using Analysis of Functional NeuroImages (AFNI; RRID: SCR_005927) (*Cox, 1996*). Nuisance variables, including six head-movement parameters, and white matter and cerebrospinal signals, were regressed out using multiple linear regression methods (FSL's *fsl_glm*). Finally, the images were intensity normalized, band-pass filtered (0.008–0.01 Hz, AFNI's *3dFourier*), registered to MNI space and spatially smoothed (Gaussian kernel, sigma = 3).

### Functional connectivity-based parcellation

To perform the functional connectivity-based parcellation, we manually drew an ROI that included the anterior olfactory nucleus, olfactory tubercle, and piriform cortex on the MNI152 template brain, according to the Atlas of the Human Brain (*Mai et al., 2015*). The ROI was drawn onto coronal slices in the range of y = −3 to y = 15 on the MNI152_T1_1mm_brain, which was down-sampled to MNI152 2 mm space afterwards. For the comparison between parcellated clusters and anatomical

subregions, we drew ROIs delineating the primary olfactory subregions prior to running the parcellation analysis. Notably, the human primary olfactory subregions differ in relative size and shape from their rodent analogs. We followed the definition of the olfactory tubercle and anterior olfactory nucleus used in the Atlas of the Human Brain (*Mai et al., 2015*; *Ongür et al., 2003*), and their locations agree with previous human studies (*Allison, 1954*; *Crosby and Humphrey, 1941*; *Eslinger et al., 1982*). Although we used an atlas that includes particularly detailed delineations of olfactory areas, based on rigorous techniques, including data from four human brains (*Ongür et al., 2003*), some researchers have postulated that the human anterior olfactory nucleus is part of the olfactory bulb (*Daniel and Hawkes, 1992*; *Hyman et al., 1991*) and that the human olfactory tubercle is actually the anterior perforated substance (*Daniel and Hawkes, 1992*; *Hyman et al., 1991*). However, these discrepencies are mainly terminology-based, as the different studies refer to the same anatomical structures.

We next parcellated the ROI into subregions based on their connectivity patterns with the rest of the brain. To do so, the Pearson correlation coefficient was computed between each voxel within the ROI and every other voxel in the rest of the whole brain, resulting a correlation matrix for each subject. The whole-brain mask was created using FSL's gray matter tissue prior image (avg152T1_gray.img, threshold of 100). The correlation coefficient was Fisher's z transformed and then averaged across subjects. The resulting matrix was transformed back into Pearson correlation coefficients, which were later used for parcellation analysis.

The parcellation analysis was performed using standard k-means methods, as implemented in the Matlab Statistics Toolbox (Matlab R2016b, The Mathworks Inc, Natick, MA USA; RRID:SCR_001622). The correlation between the connectivity pattern of the voxels within the ROI was used as the distance measure. Although this method is unsupervised, it requires an input of the number of clusters. Our hypothesis was that subregions of the primary olfactory areas within the ROI were separable based on their functional connectivity patterns. Thus, we chose a k value of four as the input for the algorithm. The clustering was performed on left and right hemisphere ROIs separately.

To evaluate the stability of the group-level connectivity patterns, we tested the stability of the correlation matrix using a leave-one-out method (*Kahnt et al., 2012*). This analysis tests whether the connectivity profile of individual voxels in primary olfactory cortex is similar across subjects, as required for averaging. For this, the correlation matrices were averaged across N−1 subjects, where N is the total number of subjects, and the correlation with the left-out subject was calculated for each voxel. We repeated this procedure N times and took the average of these repetitions as the final stability map.

To calculate the proportion of voxels from each parcellation cluster located within each anatomical subdivision of primary olfactory subregions, we first used the Atlas of the Human Brain (*Mai et al., 2015* to outline anatomical ROIs of anterior olfactory nucleus, olfactory tubercle, frontal piriform cortex and temporal piriform cortex prior to performing the parcellation analysis. We then determined the proportion of voxels from each parcellated subdivision that were located within each ROI. The significance of the proportion was tested using a permutation method. In each permutation, we shuffled the labels of the voxels within the ROI and re-calculated the proportion number. This procedure was repeated 10,000 times, resulting in a null distribution of the proportion of voxels of each parcellated subdivision within each anatomical subregion. The mean and standard deviation of this distribution was computed by norm line fitting (Matlab's *normfit*). A z score of the real proportion was computed by subtracting the average and then dividing by the standard deviation.

To characterize the functional connectivity pattern of each subregion, the time series of all voxels were averaged for each ROI. The Pearson correlation between the average time series and every other voxel in the brain was calculated for each subject and Fisher's z transformed. Finally, random effects analysis of the functional connectivity maps was performed using one-sample t tests (FSL's *randomize*, 10,000 permutations). Multiple comparisons were corrected using the TFCE method (*Smith and Nichols, 2009*). TFCE allows the identification of clusters in data sets without defining the clusters in a binary way. The output is a weighted sum of the local clustered signal. TCFE has been shown to produce results with better stability than other methods (*Smith and Nichols, 2009*). For functional connectivity network analysis, we focused on positive functional connectivity only, since the mechanisms of negative functional connectivity are less understood (*Murphy et al., 2009*).

To examine the functional connectivity difference between left and right subregions of the ROI, we computed a lateralization index of the functional connectivity. The lateralization index was calculated as $(Z_{left}-Z_{right})/(Z_{left} + Z_{right})$, where $Z_{left}$ and $Z_{right}$ represent the whole-brain functional connectivity map of the left and right subregions respectively. The lateralization index was further spatially smoothed (Gaussian kernel, sigma = 3). A one-sample t test analysis of the lateralization index was conducted using FSL's *randomize* (10,000 permutations) and multiple comparisons were corrected using the TFCE method. Because we found no statistical difference between the maps across hemispheres, we combined the corresponding left and right primary olfactory cortex clusters, and performed all subsequent analyses on these hemispherically-combined seed regions.

For the replication of our parcellation analysis, we performed k-means clustering on subjects from an independent resting-state fMRI dataset that was published elsewhere (*Kahnt and Tobler, 2017*). In brief, 6 min of resting fMRI data were collected from fifty-three healthy subjects on a Philips Achieva 3T scanner (TR: 2000 ms; TE: 30 ms; voxel size: $2.75 \times 2.75 \times 3$ mm$^3$; flip angle: 90°). A high-resolution, T1-weighted MPRAGE anatomical image was acquired (TR: 8.2 ms; TE: 3.8 ms; FOV: 256 mm; voxel size: $1.0 \times 1.0 \times 1.0$ mm$^3$; 181 slices; flip angle: 8°) for each subject. All analyses performed on this data set were identical to the steps performed in our initial analysis. Full details on the acquisition parameters can be found in *Kahnt and Tobler (2017)*, but importantly, their acquisition parameters differed from ours, highlighting the replicability of our findings.

## Data availability

Source data and code have been made available via GitHub: https://github.com/zelanolab/primaryolfactorycortexparcellation.git (*Zelano Lab, 2019*; copy archived at https://github.com/elifesciences-publications/primaryolfactorycortexparcellation).

## Acknowledgements

This study was financially supported by the National Institutes on Deafness and Other Communications Disorders (NIDCD) grants R00-DC-012803 and R01-DC-016364 (to Christina Zelano) and R01-DC-015426 (to Thorsten Kahnt).

## Additional information

### Competing interests

Thorsten Kahnt: Reviewing editor, *eLife*. The other authors declare that no competing interests exist.

### Funding

| Funder | Grant reference number | Author |
|---|---|---|
| National Institute on Deafness and Other Communication Disorders | R01-DC-015426 | Thorsten Kahnt |
| National Institute on Deafness and Other Communication Disorders | R00-DC-0012803 | Christina Zelano |
| National Institute on Deafness and Other Communication Disorders | R01-DC-016364 | Christina Zelano |

The funders had no role in study design, data collection and interpretation, or the decision to submit the work for publication.

### Author contributions

Guangyu Zhou, Conceptualization, Data curation, Formal analysis, Investigation, Visualization, Methodology, Writing—original draft, Writing—review and editing; Gregory Lane, Data curation, Visualization, Writing—original draft, Project administration, Writing—review and editing; Shiloh L Cooper, Writing—review and editing; Thorsten Kahnt, Resources, Methodology, Writing—review

and editing; Christina Zelano, Conceptualization, Formal analysis, Supervision, Funding acquisition, Visualization, Methodology, Writing—original draft, Writing—review and editing

### Author ORCIDs

Guangyu Zhou (iD) https://orcid.org/0000-0003-0897-6465
Gregory Lane (iD) https://orcid.org/0000-0003-3101-0722
Shiloh L Cooper (iD) https://orcid.org/0000-0002-0801-6352
Thorsten Kahnt (iD) https://orcid.org/0000-0002-3575-2670

### Ethics

Human subjects: This study was approved by Northwestern University's Institutional Review Board. All participants gave their written consent before the experiment. The experiment was conducted according to the principles of the Declaration of Helsinki. Protocol #STU00201746.

### Decision letter and Author response

Decision letter https://doi.org/10.7554/eLife.47177.024
Author response https://doi.org/10.7554/eLife.47177.025

## Additional files

### Supplementary files

• Transparent reporting form
DOI: https://doi.org/10.7554/eLife.47177.022

### Data availability

Source data have been provided for each figure and table. Source data and code have been made available via GitHub: https://github.com/zelanolab/primaryolfactorycortexparcellation.git (copy archived at https://github.com/elifesciences-publications/primaryolfactorycortexparcellation).

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
