## [Decision Letter]

Thank you for submitting your article "Characterizing functional pathways of the human olfactory system" for consideration by *eLife*. Your article has been reviewed by three peer reviewers, and the evaluation has been overseen by a guest Reviewing Editor and Joshua Gold as the Senior Editor. The following individual involved in review of your submission has agreed to reveal his identity: Jonas Olofsson (Reviewer #1).

The reviewers have discussed the reviews with one another and the Reviewing Editor has drafted this decision to help you prepare a revised submission.

Summary:

The majority of our knowledge regarding primary olfactory cortex anatomy comes from rodents and the field needs more human data. This study uses brain imaging to add additional functional annotation to the human primary olfactory cortex.

Essential revisions:

Several reviewers asked for a more thorough review of the previous literature using fMRI to measure connectivity. Please revise the manuscript to place this study in context of previous work.

Additionally, the reviewers raised concerns and made a number of suggestions regarding technical aspects, lateralization, and addressing individual variability. Please see below for the individual points and address them as much as possible.

*Reviewer #1:*

The manuscript by Lane et al. describes an experiment whereby functional Magnetic Resonance Imaging connectivity analysis was used to parcellate the human primary olfactory cortex into separate connective pathways. Results suggest four distinct pathways emanating from primary areas (those receiving direct input from the olfactory bulb).

This manuscript is well written, especially considering the rodent neuroanatomical work which is covered in detail, and the results, which are clearly explained. Importantly, the study addresses a key issue in human olfaction, that of the presumed complementary roles of primary olfactory areas and how to define them. Prior work has used seed regions in the piriform, and sometimes even orbitofrontal, cortex to study the connectivity patterns of the olfactory system. The current study provides a more in-depth view, and will facilitate future comparisons within the olfactory network and across sensory systems. The independent validation dataset strengthens the authors' conclusions.

Overall, the study will provide a nice addition to the literature on human olfactory system connectivity, especially since the literature is currently very limited. I have a few issues for the authors to address, these issues are listed below.

The manuscript does not cite the small number of prior studies on fMRI connectivity of the olfactory cortex and it does not adequately describe how the present approach is different from that of prior work of the same kind.

Please motivate your definition of primary olfactory cortex, given the notion that the olfactory bulb is by some considered primary cortex.

Although the methods appear later in the text, there should be an early citation or mentioning of the dataset and its resting-state methodology; only the validation dataset is mentioned in the early stages of the main text.

Discuss the commonly observed problem of poor signal-to-noise ratio in olfactory regions and how the authors avoided the risk of biasing their results due to poor/variable SNR.

You found a mean correlation coefficient of 0.19 (SE 0.0041), how does that compare with prior applications of this method on similar data sets (e.g. Kahnt et al., 2012)? Please provide support for the claim that there is good similarity.

It is interesting to note that AON has a stronger lateralization pattern. Why do you think that is the case, and what are the possible implications? Does it imply a specific functional role or might it be an artifact, or evidence of less well-defined network memberships?

Please provide references for the olfactory interactions with the language regions, as is hypothesized in Discussion.

Regarding the framework, is might be noted that the shared connectivity with the OFC might explain why it is so reliably elicited even in basic/passive olfactory tasks; all primary regions project to it. Furthermore, regarding the connectivity patterns of the frontal piriform to regions responsible for facial movements, might this include sniffing?

It is a bit of a stretch to state that there is a "large body of evidence" suggesting olfaction is lateralized to the right side. The literature is in fact quite inconsistent. I suggest tempering this claim and also adding another more recent supporting citation.

*Reviewer #2:*

The conclusion appears somewhat weak: "… Our results provide important insight into the functional and anatomical organization of the human olfactory system…" Please be more detailed in terms of the significance of these findings and the consequences of it. Do we have to look at olfactory information processing differently from today on? Please speculate: Does this demonstration of parallel processing of olfactory information explain the vulnerability in aging? Or the decrease of olfactory functioning in depression? Other consequences?

Please compare the parallel processing n the olfactory system to other systems? How is that in taste? Touch?

I was always under the assumption that the AON in humans is part of the olfactory bulb (see Daniel and Hawkes, 1992; Hyman, Arriagada and Van Hoesen, 1991). Maybe you can discuss these discrepancies, too.

The authors talk about the olfactory tubercle. This is despite the anatomical impression that there is no bump in this vicinity. In fact the anterior perforated substance has been proposed as its human homologue (see Freiherr, Wiesmann, Witt in Welge-Luessen, Hummel (eds): 2014, Management of Smell and Taste Disorders; Theme, Stuttgart). Please comment.

It would be extremely valuable for the reader if the authors came up with a sketch that graphically summarizes the results in terms of the strength of the connectivity between the various regions.

Although difficult to do (and although the authors already pay a lot of attention to that), it would be excellent if the authors could try to separate more clearly between research that comes from animals and research on humans.

*Reviewer #3:*

Odor processing starts with transduction in the olfactory epithelium and continues with treatment in the olfactory bulb and cortical processing in primary and so-called secondary areas. In mammals, we have very good understanding of the anatomy and the functionality of these areas. In contrast, the functional neuroanatomy of human olfaction remains poorly understood. In this paper, Zhou and colleagues provide a significant contribution to this field by highlighting that the human primary olfactory cortex is clustered into sub-regions that anatomically corresponded to the anterior olfactory nucleus, olfactory tubercle and frontal and temporal piriform cortices. Furthermore, a dissociation in the whole-brain functional connectivity patterns was observed across the different sub-areas of the primary olfactory cortex suggesting the existence of distinct parallel processing pathways. The paper is very well written and the method used are innovative in the field of olfaction (data-driven connectivity-based parcellation techniques). My comments are below.

- Subsection “Parcellation results across hemispheres and k values”: Why it is important to combine data from both hemispheres? Asymmetry in human olfactory processing has been documented, especially by the Zatorre group in the early 90's and more recently in different studies. I would like to see how not merging the 2 hemispheres change the findings.

- The authors mentioned “Note that additional primary olfactory areas, including anterior cortical and peri- amygdala areas and entorhinal cortex (Allison, 1954; Carmichael et al., 1994; Eslinger et al., 1982; Gonçalves Pereira et al., 2005; Zatorre et al., 1992), were not included in our ROI because the exact location of olfactory afferents into these areas is poorly understood.”. I am wondering whether some areas included in the ROI (tubercule, AON), are less “poorly understood” than entorhinal cortex or amygdala in humans. Could the authors provide some more rationales for excluding these areas from the ROI?

In the same line, the authors wrote: “Interestingly, connectivity clusters were generally more extensive in the right hemisphere compared to the left hemisphere, supporting a large body of evidence suggesting that olfactory processing is lateralized to the right side (Zatorre et al., 1992).” So, why not considering the primary olfactory cortices separately from the right and the left hemispheres? There is much to gain in showing these data.

- The authors stated that “We then estimated the whole-brain functional connectivity profile of each voxel within the ROI by computing the Pearson correlation coefficient between the resting-state fMRI time-series of a given voxel and that of every other voxel in the rest of the brain. This resulted in subject-wise connectivity matrices, which were then averaged across all subjects.” Does this mean that individual variability is not considered in the rest of the analysis? If yes, how the authors could be sure that the values in the averaged matrix are not due to a sub-sample of subjects?

- In the same line, the authors wrote “To examine the similarity between the individual functional connectivity matrices, we computed a histogram of correlation values between individual matrixes and the groups matrix using a leave-one-out method (Kahnt et al., 2012). The results indicate a good similarity between the group-level and individual functional connectivity patterns (mean correlation coefficient: 0.19, standard error: 0.0041, Figure 2A).”. I can understand that the notion of similarity is relative, but here the averaged r-value is less than 0.20 which means that half of the data fall within a range of 0 to 0.20. I may not understand properly this analysis but to my view it seems that these r-values are low. Could you comment on that? Moreover, the authors must mention the significativity of these correlative analyses. Even if significant, there is a need to detail how many data-points were computed in these analyses and how such low r-values can be interpreted?

- Participants were “instructed to look at a white fixation cross on a black background and breath in-out through their nose.” As the authors know, and they are expert in the field, respiration can alter odor processing. Is there any effect of Nasal breathing on the brain imaging findings? If nasal respiration was measured, in addition to motion correction, the authors should correct for nasal breathing activity.

---

## [Author Response]

Reviewer #1:

[…] Overall, the study will provide a nice addition to the literature on human olfactory system connectivity, especially since the literature is currently very limited. I have a few issues for the authors to address, these issues are listed below.The manuscript does not cite the small number of prior studies on fMRI connectivity of the olfactory cortex and it does not adequately describe how the present approach is different from that of prior work of the same kind.

This is a very important point, and we are grateful for being made aware of having missed important previous work. We have modified the Introduction to include a discussion of the following studies. We would be grateful if you could bring our attention to any additional studies we might have missed.

We now include a discussion of the following:

Banks et al., 2016; Caffo et al., 2010; Cecchetto et al., 2019; Fjaeldstad et al., 2017; Karunanayaka et al., 2014.; Karunanayaka, Tobia and Yang, 2017; Killgore et al., 2013; Kiparizoska and Ikuta2017; Kollndorfer et al., 2015; Krusemark and Li, 2012; Nigri et al., 2013; Sreenivasan et al., 2017; Sunwoo et al., 2015; Wang et al.,2010; Wang et al., 2005.; Wang et al., 2015.

Please motivate your definition of primary olfactory cortex, given the notion that the olfactory bulb is by some considered primary cortex.

We agree with you that the olfactory bulb is considered by some to be “primary olfactory cortex”. However, we feel that the “classic” and dominant terminology in the field still uses the term “primary olfactory cortex” to refer to structures that receive direct input from the olfactory bulb. For example, in *Olfactory Cortex: Comparative Anatomy*, Illig and Wilson discuss the concept that the olfactory bulb may perform primary sensory cortical functions, but still mention that piriform cortex is typically referred to as primary olfactory cortex. Most book chapters, reviews and studies still refer to primary olfactory cortex as brain areas that receive direct bulb projections. We feel it would be confusing to introduce new terminology in this manuscript, and might take away from the primary goal of our study, which was to provide a quantitative characterization of the parallel processing pathways of the human olfactory system. In humans, these parallel pathways occur *downstream* of the olfactory bulb. Thus, the important point here is not so much whether we are characterizing primary versus secondary olfactory cortex, but that we are characterizing the parallel pathways, all of which receive direct projections from the olfactory bulb. That said, your point is highly relevant for this paper, and reflects an important emerging movement in the field towards the idea that the classic terminology is flawed. Further, our initial phrasing of this in the Introduction may have mis-characterized the Wilson, 2009 reference. We have clarified these points in the second paragraph of the Introduction. We now initially refer to these areas as the cortical targets of olfactory bulb projections, and we refer readers to papers that discuss the accuracy of their being collectively termed “primary olfactory cortex”.

Although the methods appear later in the text, there should be an early citation or mentioning of the dataset and its resting-state methodology; only the validation dataset is mentioned in the early stages of the main text.

Very good point, thank you. This will make the paper read more clearly. We have added details about the dataset into the beginning of the Results section: “Twenty-five subjects (average ± standard error age: 25.5 ± 1.2 years; 14 female) underwent a 10-minute resting-state fMRI scan.”

Discuss the commonly observed problem of poor signal-to-noise ratio in olfactory regions and how the authors avoided the risk of biasing their results due to poor/variable SNR.

This is a helpful suggestion, and we have added a discussion of this to the Materials and methods as below. We have also added a supplementary figure showing the mean activation map for each subject, in order to show the data transparently.

“To reduce the distortion and improve the signal-to-noise ratio in the primary olfactory and orbitofrontal areas, the slice orientation was set to approximately 30° from AC-PC line (Deichmann et al., 2003). These acquisition parameters resulted in robust signals within orbitofrontal and olfactory areas in each subject (Figure 2—figure supplement 2). Further, our finding of strong connectivity between orbitofrontal cortex and all primary olfactory cortical areas suggests good orbitofrontal coverage.”

You found a mean correlation coefficient of 0.19 (SE 0.0041), how does that compare with prior applications of this method on similar data sets (e.g. Kahnt et al. 2012)? Please provide support for the claim that there is good similarity.

This is an important question, thank you for raising it. One of the other reviewers also asked about this R-value, further confirming that it’s important for us to clarify this analysis. The purpose of this leave-one-out analysis is only to estimate how similar (or stable) connectivity profiles of individual primary olfactory cortex voxels are across participants, as a prerequisite for averaging the connectivity matrices across participants. The overlap observed here was slightly lower than what Kahnt et al., 2012, observed in orbitofrontal cortex (r~0.3) using a different data set. However, this value cannot be reasonably compared across different studies because it depends on many study-specific factors including voxel size, the size of the regions of interest, and the number of data points used to compute the correlations. The important point to note about this histogram is that the similarity of connectivity patterns is above zero in all voxels in primary olfactory cortex. We have clarified this point in the second paragraph of the subsection “Parcellation of human primary olfactory cortex”.

It is interesting to note that AON has a stronger lateralization pattern. Why do you think that is the case, and what are the possible implications? Does it imply a specific functional role or might it be an artifact, or evidence of less well-defined network memberships?

We fully agree, this is a very interesting result and we should provide additional comments on this. We do not believe the finding is an artifact, and we now include, in the Results section, some discussion of this finding that ties it in with existing literature.

There are some interesting rodent findings that support the idea that the functional networks of the AON are more lateralized than other primary olfactory areas. For example, it has also been shown that the AON neurons can distinguish between signals from the ipsilateral and contralateral nostrils, suggesting representation of lateralized inputs in this region.

Please provide references for the olfactory interactions with the language regions, as is hypothesized in Discussion.

Thank you for this point. This comment made us realize that we also left out implications of our data on the neurocognitive limitations of olfactory language. We’ve added a discussion of this, along with references, in the Discussion.

Regarding the framework, is might be noted that the shared connectivity with the OFC might explain why it is so reliably elicited even in basic/passive olfactory tasks; all primary regions project to it. Furthermore, regarding the connectivity patterns of the frontal piriform to regions responsible for facial movements, might this include sniffing?

These are good and insightful points. We have added them to the Discussion.

It is a bit of a stretch to state that there is a "large body of evidence" suggesting olfaction is lateralized to the right side. The literature is in fact quite inconsistent. I suggest tempering this claim and also adding another more recent supporting citation.

We agree. We have toned down this language by removing

“supporting a large body of evidence suggesting that olfactory processing is lateralized to the right side”. We have also added more recent supporting citations.

Reviewer #2:

The conclusion appears somewhat weak: "…Our results provide important insight into the functional and anatomical organization of the human olfactory system..:" Please be more detailed in terms of the significance of these findings and the consequences of it. Do we have to look at olfactory information processing differently from today on?

Thank you for this suggestion, we agree. We have modified the Introduction and Discussion to better distinguish our findings from previous studies, and to clarify the reasons why we believe the study is important. Your suggestion really improved the Introduction, thank you. See the new additions in the fifth paragraph of the Introduction, and in the first paragraph of the Discussion.

Please speculate: Does this demonstration of parallel processing of olfactory information explain the vulnerability in aging? Or the decrease of olfactory functioning in depression? Other consequences?

Great points, thank you. We have added a discussion of these ideas to the Discussion, which has improved this part of the manuscript substantially:

“Our findings provide a detailed description of the particular brain areas which exhibit unique connectivity with each individual primary olfactory subregion. […] For example, olfactory structures that form networks with brain areas that are implicated in particular pathologies likely perform critical olfactory sensory functions (the olfactory tubercle is connected to areas involved in depression, see Croy and Hummel, 2017, and the temporal piriform cortex is connected to areas involved in primary progressive aphasia, see Olofsson et al., 2013).”

Please compare the parallel processing n the olfactory system to other systems? How is that in taste? Touch?

This is a good suggestion. We now highlight that the organization of the olfactory system differs from other sensory modalities. We attempted a detailed review of the organization of other sensory systems, however, in the end we felt that it disrupted the flow of the manuscript and we removed it. If the reviewer feels strongly that it should be included, we will add it back in.

I was always under the assumption that the AON in humans is part of the olfactory bulb (see Daniel and Hawkes, 1992; Hyman, Arriagada and Van Hoesen, 1991). Maybe you can discuss these discrepancies, too.

This is an important point, which we should have discussed more in the manuscript. The human AON certainly seems to differ anatomically from the rodent AON. In humans, it is often referred to as the “retrobulbar region”, but it has been demonstrated to be a true cortical structure, equivalent to paleocortical structures in rodents (Stephan H: Allocortex. In Bargmann W, editor: “Handbuch der mikroskopischen Anatomie des Menschen”, vol 4, Berlin and New York. Part 9, 1975, Springer Verlag, pp 1–998). In fact, the term anterior olfactory nucleus has been suggested to be misleading, given that it is a true cortical structure in humans (Zilles and Amunts, Architecture of the Cerebral Cortex, 3rd Edition, 2012). A recent study by Ongur et al. performed a detailed mapping of the human AON based on data from 5 post-mortem human brains (Ongur, 2003), which was updated in their Atlas of the Human Brain (Mai et al., 2016). Their study showed that the human AON is extensive, extending more than 11mm in the anterior-posterior direction, from MNI position y=14.22mm to MNI position y=2.53mm, as detailed in the 4th Edition of the Atlas of the Human Brain (Mai et al., 2016). Numerous studies (most of which focus on Parkinson’s Disease pathology) have characterized the human anterior olfactory nucleus into several subdivisions other than the bulbar part. Just as one example, a study by Ubeda-Banon et al. in 2010 divided the human AON into the following different parts: bulbar, intrapeduncular, retrobulbar, cortical anterior and cortical posterior subregions. The two cortical subregions could be further divided into medial and lateral components on either side of the olfactory tract. See Author response image 1, taken from this study, which shows the size and extent of the human AON. Some other studies including subdivisions of the AON beyond the olfactory bulb include:

- Pearce, Hawkes and Daniel, 1995, Movement Disorders;

- Price JL (1990) Olfactory system. In: Paxinos G (ed) The human nervous system. Academic Press, San Diego, pp 979–998;

- Hoogland PV, Huisman E (1999) Tyrosine hydroxylase immunoreactive structures in the aged human olfactory bulb and olfactory peduncle. J Chem Neuroanat 17:153–161; -Mai JK, Paxinos G, Voss T (2008) Atlas of the human brain. Elsevier, New York;

- Price JL (1990) Olfactory system. In: Paxinos G (ed) The human nervous system. Academic Press, San Diego, pp 979–998;

- Braak, H., Del Tredici, K., Rüb, U., de Vos, R. A., Jansen Steur, E. N., and Braak, E. (2003a). Staging of brain pathology related to sporadic Parkinson’s disease. Neurobiol. Aging 24, 197–211;

- Sengoku R, Saito Y, Ikemura M, Hatsuta H, Sakiyama Y, Kanemaru K, Arai T, Sawabe M, Tanaka N, Mochizuki H, Inoue K, Murayama S (2008) Incidence and extent of Lewy body-related α-synucleinopathy in aging human olfactory bulb. J Neuropathol Exp Neurol 67:1072–1083;

- Ubeda-Banon et al., 2017.

In our study, we did not distinguish the subdivisions of the human AON, though this could be of high interest for future studies.

We now include a brief description of these points in the Materials and methods subsection “Functional connectivity-based parcellation”.

The authors talk about the olfactory tubercle. This is despite the anatomical impression that there is no bump in this vicinity. In fact the anterior perforated substance has been proposed as its human homologue (see Freiherr, Wiesmann, Witt in Welge-Luessen, Hummel (eds): 2014, Management of Smell and Taste Disorders; Theme, Stuttgart). Please comment.

We agree that the anatomy of the rodent and human olfactory tubercle differs. Similar to the AON, the human tubercle differs in relative size and shape from the rodent homologue. We followed the definition of the olfactory tubercle used in the Atlas of the Human Brain (Mai et al., 2015; Öngür et al., 2003), as mentioned in our previous response above. The location of the OT in this atlas agrees with previous studies that we cite in the paper (Allison, 1954; Crosby and Humphrey, 1941; Eslinger et al., 1982). It does appear that the currently accepted location of the human olfactory tubercle overlaps with the location of the anterior perforated substance. We have added mention of this to our Materials and methods, and cite the book mentioned.

It would be extremely valuable for the reader if the authors came up with a sketch that graphically summarizes the results in terms of the strength of the connectivity between the various regions.

This is a great point. Our connectivity maps did not contain information about the relative strength of the different connections. We have modified our figures to include t-values rather than simple binarized values. These new maps, that contain information about the strength of the connectivity, have been inserted to replace Figure 4.

Although difficult to do (and although the authors already pay a lot of attention to that), it would be excellent if the authors could try to separate more clearly between research that comes from animals and research on humans.

We agree this is an important point, and a common issue in human olfactory studies/manuscripts, since the vast majority of studies are conducted in rodents. We have made efforts wherever possible to make this distinction more clear.

Reviewer #3:

[…] - Subsection “Parcellation results across hemispheres and k values”: Why it is important to combine data from both hemispheres? Asymmetry in human olfactory processing has been documented, especially by the Zatorre group in the early 90's and more recently in different studies. I would like to see how not merging the 2 hemispheres change the findings.

This is an excellent point. We will clarify our thinking here and in the text.

Not merging the 2 hemispheres does not change the findings in a statistically relevant way.

We merged hemispheres for three main reasons. First, there are no significant differences between hemispheres, based on a statistical comparison of the connectivity maps. Second, given this lack of statistical difference across hemispheres, to show each hemisphere for each subregion would add additional information without additional meaning, to the manuscript, while making it denser and more difficult to interpret for the reader. Third, and most important, to show such data would risk misleading the reader and could lead to misinterpretation of our results.

Because of these concerns, and particularly the concern that readers could mis-cite our paper as showing differences across hemispheres when we found none statistically, we feel strongly that we should not add supplementary figures showing the single hemisphere maps. However, if the reviewer feels very strongly that we should add this figure, we will create a supplementary figure showing these maps.

Based on this helpful comment, we have added clarification of this to the Materials and methods subsection “Functional connectivity-based parcellation”.

- The authors mentioned “Note that additional primary olfactory areas, including anterior cortical and peri- amygdala areas and entorhinal cortex (Allison, 1954; Carmichael et al., 1994; Eslinger et al., 1982; Gonçalves Pereira et al., 2005; Zatorre et al., 1992), were not included in our ROI because the exact location of olfactory afferents into these areas is poorly understood.”. I am wondering whether some areas included in the ROI (tubercule, AON), are less “poorly understood” than entorhinal cortex or amygdala in humans. Could the authors provide some more rationales for excluding these areas from the ROI?

Thank you for this insightful comment. It is a good point, and we did not adequately explain our rationale in the manuscript.

The AON and OT are better understood *anatomically* (though we agree, not functionally) than the olfactory inputs to the amygdala and entorhinal cortex. For example, a recent histological study by Goncalves et al. found that the borders of the subdivisions of the amygdala and entorhinal cortex that receive direct bulb projections are not discernable in T1 images (Gonçalves Pereira et al., 2005). Furthermore, we were unable to find a human brain atlas that clearly outlines olfactory amygdala and entorhinal areas receiving bulb input, whereas numerous studies and atlases do include delineation of the olfactory tubercle and AON. We have modified the manuscript to include this reference where we discuss this point in the first paragraph of the Results subsection “Parcellation of human primary olfactory cortex”.

In the same line, the authors wrote: “Interestingly, connectivity clusters were generally more extensive in the right hemisphere compared to the left hemisphere, supporting a large body of evidence suggesting that olfactory processing is lateralized to the right side (Zatorre et al., 1992).” So, why not considering the primary olfactory cortices separately from the right and the left hemispheres? There is much to gain in showing these data.

Thank you for this comment. Interestingly, we found that across the whole-brain, connectivity was stronger between *all (both left and right)* primary olfactory areas and clusters in the right hemisphere. Importantly, this whole-brain hemispheric effect did not differ between primary olfactory cortical hemispheres—that is, both the right and left primary olfactory cortices exhibited increased right-sided whole-brain connectivity. This is a subtle distinction, and is shown in the figure displaying the common connectivity pattern (Figure 5). We clarified this in the subsection “Functional connectivity common to all subregions”.

- The authors stated that “We then estimated the whole-brain functional connectivity profile of each voxel within the ROI by computing the Pearson correlation coefficient between the resting-state fMRI time-series of a given voxel and that of every other voxel in the rest of the brain. This resulted in subject-wise connectivity matrices, which were then averaged across all subjects.” Does this mean that individual variability is not considered in the rest of the analysis? If yes, how the authors could be sure that the values in the averaged matrix are not due to a sub-sample of subjects?

This is a very important point, thank you for bringing this up. We computed the connectivity maps on an individual basis, but averaged the resulting connectivity matrices to obtain a group-connectivity map, as done previously (Kahnt et al., 2012; Kahnt and Tobler, 2017). For this, we first determined whether the connectivity structure of individual primary olfactory cortex voxels was indeed comparable across subjects. This was done using a leave-one-out analysis, which showed that there was substantial overlap, as required for averaging across subjects. We have clarified this in the second paragraph of the subsection “Parcellation of human primary olfactory cortex”, where we mention averaging across subjects.

- In the same line, the authors wrote “To examine the similarity between the individual functional connectivity matrices, we computed a histogram of correlation values between individual matrixes and the groups matrix using a leave-one-out method (Kahnt et al., 2012). The results indicate a good similarity between the group-level and individual functional connectivity patterns (mean correlation coefficient: 0.19, standard error: 0.0041, Figure 2A).”. I can understand that the notion of similarity is relative, but here the averaged r-value is less than 0.20 which means that half of the data fall within a range of 0 to 0.20. I may not understand properly this analysis but to my view it seems that these r-values are low. Could you comment on that? Moreover, the authors must mention the significativity of these correlative analyses. Even if significant, there is a need to detail how many data-points were computed in these analyses and how such low r-values can be interpreted?

These are very important points for our study, thank you for these comments. The purpose of this leave-one-out analysis was solely to determine whether we could reasonably average across the connectivity maps of individual subjects. Similar to previous studies (e.g., Kahnt et al., 2012) we expect this R value to be relatively small. Moreover, this value will likely differ across studies because it depends on study-specific differences in ROI sizes and voxel size. However, as can be seen in the histogram, most of these correlations were above zero, and because they are computed across rest-of-the-brain voxels (N=170,660), also statistically significant (R values larger than 0.0088 are significant at p<0.05, Bonferroni corrected for the number of voxels in the primary olfactory cortex). We have clarified these points in the second paragraph of the subsection “Parcellation of human primary olfactory cortex”.

- Participants were “instructed to look at a white fixation cross on a black background and breath in-out through their nose.” As the authors know, and they are expert in the field, respiration can alter odor processing. Is there any effect of Nasal breathing on the brain imaging findings? If nasal respiration was measured, in addition to motion correction, the authors should correct for nasal breathing activity.

Excellent point. In this study, we focused on low-frequency (< 0.1 Hz) oscillations, which are below the human respiratory frequency (typically from 0.15 – 0.3 Hz). Although there’s increasing interest in studying higher frequency brain activity using fMRI, the resting functional connectivity profile of high frequency oscillations and their relationship with respiration is not well understood. Our previous study has shown that nasal breathing can modulate local brain activity in a way that is different from oral breathing, and it would be interesting to examine whether functional networks are distinct during those two different breathing routines in future studies.